

# Limited protection of macro-aggregate occluded organic carbon in Siberian steppe soils

Norbert Bischoff[1], Robert Mikutta[2], Olga Shibistova[1,3], Alexander Puzanov[4], Marina Silanteva[5], Anna Grebennikova[5], Roland Fuß[6], Georg Guggenberger[1,3]

[1]Institute of Soil Science, Leibniz Universität Hannover, Hannover, 30419, Germany

[2]Soil Science and Soil Protection, Martin-Luther University Halle-Wittenberg, Halle, 06120, Germany

[3]VN Sukachev Institute of Forest, Siberian Branch of the Russian Academy of Sciences, Krasnoyarsk, 660036, Russian Federation

[4]Institute for Water and Environmental Problems, Siberian Branch of the Russian Academy of Sciences, Barnaul, 656038, Russian Federation

[5]Faculty of Biology, Altai State University, Barnaul, 656049, Russian Federation

[6]Institute of Climate-Smart Agriculture, Johann Heinrich von Thünen Institute, Braunschweig, 38116, Germany

*Correspondence to*: Norbert Bischoff (bischoff@ifbk.uni-hannover.de)





**Abstract.**

Macro-aggregates especially in agricultural steppe soils are supposed to play a vital role for soil organic carbon (OC) stabilization at a decadal time scale. While most research on soil OC stabilization in steppes focused on North American prairie soils of the Great Plains with information mainly provided by short-term incubation experiments, little is known about the agricultural steppes in south-western Siberia, though they belong to the greatest conversion areas in the world and occupy an area larger than that in the Great Plains. To quantify the proportion of macro-aggregate protected OC under different land-use and as function of land-use duration and intensity in Siberian steppe soils, we determined OC mineralization rates of intact (250–2000 μm) and crushed (<250 μm) macro-aggregates in long-term incubations over 401 days (20°C; 60% water holding capacity) along two agricultural chronosequences in the Siberian Kulunda steppe. Additionally we incubated bulk soil (<2000 μm) to determine the effect of land-use change (LUC) and subsequent agricultural use on a fast and a slow soil OC pool (labile *vs.* more stable OC), as derived from fitting exponential decay models to incubation data. We hypothesized that (i) macro-aggregate crushing leads to increased OC mineralization due to an increasing microbial accessibility of a previously occluded labile macro-aggregate OC fraction, and (ii) bulk soil OC mineralization rates and the size of the fast OC pool are higher in pasture than in arable soils with decreasing bulk soil OC mineralization rates and size of the fast OC pool as land-use duration and intensity increase. Against our hypothesis, OC mineralization rates of crushed macro-aggregates were similar to those of intact macro-aggregates under all land-use regimes. Macro-aggregate protected OC was almost absent and accounted for <1% of the total macro-aggregate OC content and to maximally 8 ± 4% of mineralized OC. In accordance to our second hypothesis, highest bulk soil OC mineralization rates and sizes of the fast OC pool were determined under pasture, but mineralization rates and pool sizes were unaffected by the duration and intensity of land-use. However, mean residence times of the fast and slow OC pool tended to become shorter along one chronosequence. We conclude, that the tillage-induced break-down of macro-aggregates has not reduced the OC contents in the soils under study. The decline of OC after LUC is probably attributed to the faster soil OC turnover under arable land as compared to pasture at a reduced plant residue input.



## Introduction

Steppe soils comprise about 7% of the terrestrial soil organic carbon (OC) storage down to 1m (Calculation see supplementary material) and cover about 885 million ha worldwide (FAO, 2001). As they are rich in organic matter (OM) and well-suited for agriculture they encompass about 14% of agricultural land globally (FAO, 2013). Intensive management

of steppe soils reduced their OC stocks significantly, with estimated OC losses between 24 and 40% associated with conversion of grassland to cropland (Beniston et al., 2014; Mikhailova et al., 2000; Rodionov et al., 1998; VandenBygaart et al., 2003). As the stabilization of OC in agricultural steppe soils is crucial for maintaining soil fertility and to reduce the emission of $CO_2$ to the atmosphere, further insights into the processes that govern OC stabilization in steppe soils are needed. For temperate soils chemical stabilization by formation of mineral-organic associations and physical disconnection

of OM from microorganisms by occlusion of OM in aggregates, were identified as main factors stabilizing soil OC (von Lützow et al., 2006). For dry steppe ecosystems the role of aggregation might be more decisive for OC stabilization than the one of mineral-organic associations, as the latter requires sufficient water for the formation of pedogenic minerals and the interaction of OM with mineral surfaces (Kleber et al., 2015).

The mean residence time of aggregate-occluded OC ranges from decades to several hundreds of years (Six et al.,

2002). Tisdall and Oades (1982) proposed a concept in which aggregates are structured hierarchically with respect to their size and binding agents. According to this aggregate hierarchy concept, free primary particles or silt-sized aggregates (<20 μm) are bound together to micro-aggregates (<250 μm) by persistent binding agents, e.g. humified OM, polyvalent metal cations or oxides. The micro-aggregates, in turn, are linked together to form larger macro-aggregates (>250 μm) by temporary (e.g. fungal hyphae, roots) or transient binding agents (e.g. microbial and plant-derived polysaccharides). Due to

the hierarchical order of aggregate structure and the different persistence of the involved binding agents, macro-aggregates are less stable and more vulnerable to soil management than micro-aggregates (Tisdall and Oades, 1982). Accordingly, Six et al. (2000b) showed that macro-aggregates disintegrated more readily upon disturbance than micro-aggregates, particularly in soils with increasing cultivation intensity. By that, macro-aggregate occluded OC becomes available to microbial decomposition, hence, this fraction is supposed to play an important role for the decline of soil OC in intensively managed

steppe soils (Cambardella and Elliott, 1993, 1994; Elliott, 1986).

One way to quantify the proportion of macro-aggregate protected soil OC is to compare mineralization rates from intact and crushed macro-aggregates. Previous studies found an increase of soil OC mineralization after macro-aggregate crushing (Beare et al., 1994; Bossuyt et al., 2002; Elliott, 1986; Gupta and Germida, 1988; Pulleman and Marinissen, 2004), though not all studies revealed consistent results (Garcia-Oliva et al., 2004; Goebel et al., 2009; Plante et al., 2009; Tian et

al., 2015). Moreover, OC mineralization after macro-aggregate crushing differed also with respect to land-use. Pulleman and Marinissen (2004) found larger mineralization after crushing of macro-aggregates in croplands than in grasslands and ascribed this to the physicogenic nature of macro-aggregates in arable soils, which have smaller pore sizes than biogenic macro-aggregates in grasslands, and therefore larger protection capacity. Also Elliott (1986) observed the increase of OC



mineralization with macro-aggregate crushing to be more pronounced in arable than in grassland soils, while Gupta and Germida (1988) observed the opposite effect. A shortcoming of previous studies is the short incubation period of only few weeks resulting in non-equilibrium mineralization rates, which complicates an accurate assessment of the size of the macro-aggregate protected OC fraction and its turnover time. This fact, therefore, asks for long-term incubation experiments to
address the vulnerability of macro-aggregate protected OC.

      The majority of research on OC protection in aggregates of steppe soils focused on prairie soils of the Great Plains, while little is known for Siberian steppe soils. This is surprising as the semi-arid steppe ecosystems in Siberia belong to the greatest agricultural production areas in the world with an area greater than that of the Great Plains and cover some of the most intensively managed soils globally (Frühauf, 2011). In the West Siberian Plain 420,000 km² natural steppe was
converted into cropland between 1954 and 1963 in the frame of the so-called "Virgin Lands Campaign" (Russian: *Zelina*). Conversion from grassland to cropland reduced soil OC stocks by about 31% in 0-25 cm, of which most occurred within the first years after land conversion and was associated with a decline in aggregate stability (Bischoff et al., 2016). This indicated an interrelation between aggregate stability and OC storage also in these soils. Moreover, Bischoff et al. (2016) found about 10% of OC in the studied soils was existent in particulate OM of which some is probably occluded within
aggregates. In the present study we aimed to quantify the proportion of macro-aggregate protected OC under different land-use and as function of land-use duration and intensity in Siberian steppe soils by comparing OC mineralization rates of intact (250–2000 μm) and crushed (<250 μm) macro-aggregates in long-term incubations over 401 days along two agricultural chronosequences of the south-western Siberian Kulunda steppe. In addition, bulk soil samples (<2000 μm) were incubated to determine the effect of land-use change (LUC) from pasture to arable land on a fast and a slow soil OC pool (labile *vs.* more
stable OC), as derived from fitting exponential decay models to incubation data. We hypothesized that (i) crushing of macro-aggregates leads to increased OC mineralization due to an increasing microbial accessibility of a previously occluded labile macro-aggregate OC fraction, and (ii) bulk soil OC mineralization rates and the size of the fast soil OC pool are higher in pasture than in arable soils with decreasing bulk soil OC mineralization rates and size of the fast OC pool as land-use duration and intensity increase. In this study, we refer to fractions as physically separated soil OC components (macro-
aggregate occluded soil OC), while pools refer to mathematically derived OC constituents from fitting exponential decay models to incubation data (fast and slow soil OC pool).



## Material & Methods

### Study sites and soil sampling

The Kulunda steppe is part of the Russian Federation (Altayskiy Kray) and located within the semi-arid steppes of south-western Siberia. We selected two sites in two different steppe types under different climate with soils of different textures 5 (Fig. 1). The first site is located in the forest steppe (FS) near Pankrushikha (53°44'19.53"N, 80°41'2.88"E) with a mean annual precipitation (MAP) of 368 mm and a mean annual temperature (MAT) of 1.1°C (Table 1). The second site is situated near Sidorovka (52°30'1.43"N, 80°44'41.68"E) and part of the more arid typical steppe (TS) with a MAP of 339 mm and a MAT of 2.0°C (climate data from "WorldClim" data base; Hijmans *et al.*, 2005). At each site we identified a land-use chronosequence with four plots. At FS, we also included two plots with varying land-use intensity (extensive pasture *vs.* 10 arable land with forage crops). The FS site comprised an extensive pasture (vegetation: *Festuca valesiaca - Fillipendula vulgaris - Bromopsis inermis*), an arable land with forage crops and arable land after five and ten years of cultivation (arable 5 yr, arable 10 yr). Crop rotations on the arable 5 yr and arable 10 yr included summer wheat, summer barley and peas. The soils were classified as Protocalcic Chernozems (Siltic) according to IUSS Working Group WRB (2014). The TS site consisted of four plots which were all cultivated since the 1950s (*Zelina*) but left as fallow since 1983 because of low 15 agricultural productivity. After 1983 all plots were used extensively as pasture but three of the four plots were recultivated at different points in time, allowing for a chronosequence with a 30-year old fallow (meanwhile used as pasture) and plots with one, three, and ten years arable land-use (arable 1 yr, arable 3 yr , arable 10 yr). The 30-year old fallow (pasture) is characterized by *Agropyron pectinatum*, *Bromopsis inermis* and *Artemisia glauca*. The absence of some typical steppe species like *Stipa sp.* or *Festuca sp.* pointed to the fact that the vegetation of this plot was degraded from grazing. The site 20 was located on a small hillslope with <2° inclination, where the fallow 30 yr was located at the highest point and the arable 10 yr at the base level. Though the inclination was very small, we measured larger soil OC contents in the arable 10 yr plot than in the upslope arable 1 yr and arable 3 yr plots, which we attributed to erosion. Nevertheless, we decided to include this site in our study, as chronosequences are very sparse in the study area and the possible effect of macro-aggregate crushing on soil OC mineralization, if existent, will be also evident on slightly eroded plots. Soils at the TS site were classified as 25 Protocalcic Kastanozem (Loamic). At both sites one characteristic key profile was established from 0-150 cm on the pasture plots for soil description and sampled in generic horizons. As the arable 5 yr and arable 30 yr at the FS site were >500 m distant from the other two plots, we additionally established a key profile on each of these two plots. Key profile samples were analyzed for pH, soil texture, and electrical conductivity (EC). Further, on all plots three additional soil samples (field replicates) were randomly collected in 0-10 cm depth for determination of soil OC and total nitrogen (TN) content and for 30 use in the soil incubation experiment. Geographical coordinates of all plots are summarized in Table S1.





**Sample preparation and basic soil analyses**

Soil samples were air-dried and sieved to <2 mm. Big clods were gently broken apart to pass the 2 mm sieve and all visible plant residues were removed. A subsample was dried at 105°C for 24 h to determine the residual soil water content. Another subsample was homogenized with a ball mill (Retsch MM200, Haan, Germany) and measured for OC and TN via dry

combustion with an Elementar vario MICRO cube C/N Analyzer (Elementar Analysensysteme GmbH, Hanau, Germany). Traces of inorganic carbon ($CaCO_3$-content <0.1%) were previously removed by HCl fumigation (Walthert et al., 2010). Soil pH was measured at a 1:2.5 (w:v) soil-to-water$_{deion}$ ratio after leaving the suspensions for one day to reach equilibrium, and soil EC was measured at a soil-to-water$_{deion}$ ratio of 1:5 (w:v). The texture of the soils was determined according to the standard sieve-pipette method (DIN ISO 11277, 2002).

**Aggregate crushing and incubation of soil samples**

Each of the samples from 0-10 cm was divided into three fractions: (i) bulk soil (<2000 µm), (ii) intact macro-aggregates (250–2000 µm), and (iii) crushed macro-aggregates (<250 µm). Intact macro-aggregates were isolated by gently sieving the air-dry bulk soil through a 250-µm sieve and using the fraction remaining on the sieve. A subsample from the intact macro-aggregates was crushed in a mortar and sieved again through the 250-µm sieve to obtain the fraction of crushed macro-

aggregates (<250 µm). We decided to use dry-sieved aggregates for soil incubation as wet-sieving releases soluble OM, which is bioavailable and thus a critical fraction for soil OC mineralization (Sainju, 2006). Further, microbial activity is less affected by dry-sieving than by wet-sieving (Sainju, 2006). All samples of the three fractions were divided into three analytical replicates, giving a total of 216 samples for soil incubations (8 plots x 3 field replicates x 3 fractions x 3 analytical replicates).

To determine whether the crushed macro-aggregates consisted of intact micro-aggregates or free primary particles, a subsample of crushed macro-aggregates was sieved through a 63-µm sieve and obtained fractions were imaged by a JEOL JSM-6390A scanning electron microscope (JEOL Ltd., Tokyo, Japan). Our analysis revealed that 62.1 ± 3.2% of crushed macro-aggregates still existed as large micro-aggregates (>63 µm), while 37.9 ± 3.2% were found in the fraction <63 µm, which mainly consisted of small micro-aggregates and only few free primary particles (Fig. S1).

Soil laboratory incubations were carried out under aerobic conditions in the dark, at constant temperature of 20°C and 60% of water holding capacity (WHC). An amount of 7.5 g soil sample was mixed with 12.5 g combusted (1000°C for 24 h) quartz powder (Roth, Karlsruhe, Germany; >99% pulverized, <125 µm) and filled into 120-ml glass jars. Quartz powder was used to increase the sample volume and prevent the formation of aggregates in the crushed samples. Three jars were solely filled with quartz and used as control. Soil moisture was regulated during the experiment by periodically

weighing the glass jars and adding ultrapure water. All samples were pre-incubated for 14 days and respiration measurements were subsequently taken at days 1, 3, 8, 14, 21, 28, 57, 98, 127, 196, 268, and 401 by sampling the headspace



of each jar using a syringe through a septum, which was installed in the jar lids prior to sampling. Gas samples were analyzed for $CO_2$ concentrations with a Shimadzu GC-2014 modified after Loftfield et al. (1997).

**Determination of microbial biomass**

After the laboratory incubations all samples were analyzed for microbial biomass C using the chloroform-fumigation-extraction method (Vance et al., 1987). Briefly, 6 g soil were kept at 60% WHC and weighed in duplicate into glass jars. One sample was fumigated with ethanol-free $CHCl_3$ during 24 h while the other sample was left unfumigated. Both, fumigated and unfumigated samples, were extracted with 0.5 M $K_2SO_4$ at a soil-to-solution ratio of 1:10 (w:v), shaken for 30 min, and subsequently centrifuged at 2700 g. The extracts were filtered (Whatman filter paper, ashless, Grade 42) and measured for non-purgeable organic carbon (NPOC) by a LiquiTOC (Elementar Analysensysteme GmbH, Hanau, Germany). Microbial biomass C was calculated as the difference between fumigated and unfumigated soil samples and expressed as mg C g $OC^{-1}$.

**Calculations and statistical analyses**

All data analyses were carried out in R 3.1.2 (R Core Team, 2015). To calculate cumulative respiration rates, data of $CO_2$ measurements per day was interpolated by spline interpolation for each sample (i.e. analytical replicate) separately. Cumulative respiration rates were analyzed by fitting three different exponential-decay models to the data and choosing the model with the best fit by AIC selection (Akaike Information Criterion). The first model was a first-order exponential decay model with one pool (one-pool model; Eq. 1):

$$C_{remain} = C_1 \times e^{(-k_1 \times t)} \tag{Eq. 1}$$

The second model consisted of two pools (two-pool model; Eq. 2):

$$C_{remain} = C_1 \times e^{(-k_1 \times t)} + C_2 \times e^{(-k_2 \times t)} \tag{Eq. 2}$$

The third model was an asymptotic first-order exponential decay model with two pools (asymptotic two-pool model; Eq. 3):

$$C_{remain} = C_2 + C_1 \times e^{(-k_1 \times t)} \tag{Eq. 3}$$

where $C_{remain}$ is the amount of OC remaining in the sample, $C_1$ and $C_2$ are the sizes of the fast and the slow pool, respectively, $k_1$ and $k_2$ the rate constant of the fast and the slow pool, respectively, and $t$ the time. For the majority of samples the two-pool model showed the best fit. Only for the pasture plot at FS the incubation time was too short to calculate the rate constant $k$



for the slow pool, thus the asymptotic two-pool model fitted the data best. The mean residence time (MRT) was calculated as $1/k$. The modelled parameters were used in linear mixed effects models (package lme4; Bates et al., 2012) to test for significant differences between soil fractions within plots, accounting for the nested structure of sampling by using the field replicates within each plot as random effects. Moreover, random slopes were included by allowing field replicates within

each plot to have random slopes for the effect of soil fraction. Based on the linear mixed model fit, we tested whether differences of the dependent variable between soil fractions within plots were significant, including corrections for multiple comparisons (analogous to the Tukey test) with Satterthwaite degrees of freedom, using the R packages lsmeans (Lenth and Herve, 2015), lmerTest (Kuznetsova et al., 2015) and multcomp (Hothorn et al., 2008). Model assumptions were checked using residuals *vs.* fitted plots and Q-Q-plots for the residual errors and random effect estimates. The proportion of the

macro-aggregate protected OC fraction to the total macro-aggregate OC content was calculated by Eq. 4:

$$C_{macro,\,total\;aggrC} = C_{min,crushed} - \frac{1}{n}\ \sum_{i=1}^{n} C_{min,intact} \qquad\qquad\text{(Eq. 4)}$$

where $C_{macro,total\;aggrC}$ is the proportion of macro-aggregate protected OC to the total macro-aggregate OC (%), $C_{min,crushed}$ is

the proportion of OC mineralized in the crushed macro-aggregates (%) and $C_{min,intact}$ is the proportion of OC mineralized in the intact macro-aggregates (%), while $n$ is the number of analytical replicates per field replicate for the treatment of intact macro-aggregates and $i$ is the $i$th analytical replicate per field replicate. The proportion of the macro-aggregate protected OC fraction to the total mineralized OC as function of time was calculated by Eq. 5:

$$C_{macro,\,mineralizableC}\ (t) = \frac{C_{min,crushed}\ (t) - \frac{1}{n}\sum_{i=1}^{n} C_{min,intact}\ (t)}{C_{min,crushed}\ (t)} \times 100 \qquad\qquad\text{(Eq. 5)}$$

where $C_{macro,mineralizableC}\ (t)$ is the proportion (%) of macro-aggregate protected OC to the total mineralized OC at time $t$ (days). Graphs were generated using ggplot2 (Wickham, 2009). Boxplots show the median, the first and the third quartile and the whiskers extend from the box to the highest or lowest value, respectively, that is within $1.5 \times$ inter-quartile range.

Individual measurements are plotted as points.





## Results

### Soil organic carbon contents along the chronosequences

In FS, soil OC contents decreased as a result of LUC from pasture to arable land from $55 \pm 5$ mg g$^{-1}$ under extensive pasture to $39 \pm 1$ mg g$^{-1}$ and $40 \pm 2$ mg g$^{-1}$ under arable 5 yr and arable 30 yr, respectively (Table 1). Thus, increasing duration of

5 agricultural land-use caused no further decrease of soil OC contents in arable soils. C : N ratios were around 12 and slightly higher for non-arable than for arable soils. Soil OC contents in TS were smaller than in FS and did not follow the gradient over time since cultivation as the site was affected by erosion (Sect. 2.1). In TS, soil C : N ratios did not vary considerably between the plots.

### Effect of macro-aggregate crushing on the mineralization of soil organic carbon

Mass balance calculations revealed, that in both steppe types about 70% of OC was associated with macro-aggregates, indicating the importance of macro-aggregates for the OC dynamics in these soils. Organic C and TN contents did not vary considerably between intact and crushed macro-aggregates (Table S2). As is typically for soil incubations, respiration rates were higher at the beginning and decreased with increasing incubation time (Fig. S2). The amount of OC remaining in the sample during incubation was described by either two-pool or asymptotic two-pool models (Fig. 2). The variability of the

amount of OC remaining in the samples within one plot decreased with increasing time since cultivation (Fig. 2). Thus, soil samples belonging to the plots with the longest cultivation history were more similar to each other than samples from plots in more pristine state. The amount of soil OC mineralized was slightly larger in the bulk soil fraction ($<2000$ μm) than in the intact and crushed macro-aggregates in most of the studied plots, though significant differences were only observed in soils of FS ($p<0.05$, Fig. 3).

There was no significant difference in soil OC mineralization between intact and crushed macro-aggregates after 401 days of incubation in all plots under study (Fig. 2 and 3). The fraction of macro-aggregate protected OC was practically not existent and accounted for $<1\%$ of the total macro-aggregate OC content in all plots (data not shown). Furthermore, macro-aggregate crushing did not increase the size of the fast soil OC pool, which was determined by fitting exponential decay models to the incubation data (Fig. 4). Also the MRT of the fast and the slow OC pool was unaffected by macro-

aggregate crushing (Table 2). However, we could determine a small contribution of the macro-aggregate protected OC fraction to the total OC mineralization during the beginning of the incubation in seven out of eight plots, where macro-aggregate protected OC contributed to about 10% to the total mineralized OC d$^{-1}$ (Fig. 5). Cumulated over the entire incubation period, the contribution of the macro-aggregate protected OC fraction to the total OC mineralization was not existent or very small and amounted between zero and $8 \pm 4\%$ in seven out of eight plots with no clear trend with respect to

the land-use duration (Table 3). The arable 3 yr plot in TS had clearly negative values of macro-aggregate protected OC, which resulted from a lower OC mineralization in crushed than in intact macro-aggregates. For most plots, the negligible




fraction of macro-aggregate protected OC was depleted between 100 and 400 days, while the arable 30 yr in FS and the fallow 30 yr (pasture) in TS showed a constant but small (ca. 5%) mineralization rate of macro-aggregate protected OC during the complete incubation period (Fig. 5).

**Soil organic carbon mineralization along the chronosequences**

5   The bulk soil OC mineralization declined after LUC from pasture to arable land in both steppe types, but only in TS we observed also a trend of decreasing soil OC mineralization with increasing duration of land-use (Fig. 3). Likewise, the proportion of the fast soil OC pool decreased as a result of LUC, but it was unaffected by the duration or intensity of arable land-use (Fig. 4). The MRT of the fast OC pool became shorter in the course of LUC in both steppe types, but only in FS we observed also a trend towards shorter MRTs with increasing intensity and duration of land-use (Table 2). With respect to the 10  slow soil OC pool, only in FS we detected shorter MRTs due to conversion of pasture to arable land and with increasing intensity and duration of land-use, while no trend was apparent along the chronosequence in TS (Table 2). In general, the amount of soil OC mineralized was slightly larger in TS than in FS, while the differences were most pronounced between the pasture plots (Fig. 3). Remarkable was the pasture in TS, which had clearly the largest OC mineralization and proportion of the fast OC pool but, at the same time, also the highest variability (Fig. 3 and 4).

15 **Microbial biomass carbon**

The share of microbial biomass C in the total OC was similar in both steppe types and ranged between 1.5 and 4.0 mg C g$^{-1}$ OC, as indicated by the first and third quartile of the boxplots (Fig. 6). Crushing of macro-aggregates caused a small decrease of microbial biomass C, which was significant when considering all plots ($p < 0.05$), while bulk soil samples and intact macro-aggregates had similar amounts of microbial biomass C. There was no correlation between the amount of OC 20  mineralized and the share of microbial biomass C in total OC (Fig. S3). Moreover, the quantity of OC mineralization was not related to the amount of microbial biomass C per gram soil (Fig. S4).

**Discussion**

**Limited protection of macro-aggregate occluded organic carbon**

Previous studies showed higher OC mineralization following macro-aggregate crushing (e.g. Beare et al., 1994; Bossuyt et 25  al., 2002; Pulleman and Marinissen, 2004), while some studies showed no such effect (Garcia-Oliva et al., 2004; Goebel et al., 2009; Plante et al., 2009). In our study, the macro-aggregate occluded OC fraction contributed only marginally to the OC mineralization during the entire incubation (Fig. 5, Table 3). Against our hypothesis, macro-aggregate occluded OC is not protected against decomposition in the studied soils. In turn, the break-down of macro-aggregates due to soil tillage and the subsequent release of soil OC is not the reason for a decrease of soil OC contents due to soil management, as observed along 30  the chronosequence in FS (Table 1). Plante et al. (2009) suspected that the disruption treatments used in their experiments



(crushing of 2–4 mm aggregates to <0.5 mm) was insufficient to release large amounts of physically protected OC for decomposition, and that a considerable amount of OC was stabilized in micro-aggregates. Also Balesdent et al. (2000) provided some evidence that the proportion of physically protected OC is larger in micro-aggregates than in macro-aggregates. In our study, the majority of crushed macro-aggregates (62 ± 3%) consisted of micro-aggregates with 63–250 µm size (See Material & Methods), and as the OC mineralization of the crushed aggregate fraction was not enhanced, we suggest that most of the OC was stabilized in the micro-aggregates. However, micro-aggregates are less sensitive to soil tillage (Tisdall and Oades, 1982), therefore, in light of LUC-induced OC losses, the soil OC in macro-aggregates is generally considered to be more vulnerable for destabilization than OC in micro-aggregates. This could not be confirmed for the soils under study. Nevertheless, we cannot rule out that an increased macro-aggregate turnover due to agricultural management leads to a reduced formation of micro-aggregates within macro-aggregates and, as a result, to lower OC contents in arable as compared to pasture soils (Six et al., 2000a).

Only few studies determined the share of the macro-aggregate protected OC fraction in the total OC mineralization or the total macro-aggregate OC, respectively. Beare et al. (1994) showed that macro-aggregate protected OC accounted for about 1% of total aggregate OC and to about 8–23% of total mineralizable OC during 20 days of incubation. They detected a smaller macro-aggregate protected OC mineralization in more intensively managed soils. In our study, <1% of total macro-aggregate OC was stored as macro-aggregate protected OC, while this fraction accounted for max. 8 ± 4% of total OC mineralization (Table 3). Thus, our values are in the same order of magnitude as observed by Beare et al. (1994), who suggested that an increased macro-aggregate turnover in tilled soils is one reason for the small macro-aggregate protected OC fraction. According to Beare et al. (1994), the physically protected but relatively labile macro-aggregate occluded OC is released for microbial decomposition due to the frequent tillage-induced macro-aggregate break-down. As a result, macro-aggregates contain only little or no labile OC. This can be a reason in arable soils, but is unlikely in pasture soils where the macro-aggregate turnover is slower due to the absence of tillage (Six et al., 2002). In the untilled soils, therefore, other factors are probably responsible for the absence of labile macro-aggregate protected OC.

The mineralization of OC is driven by microorganisms and, thus, can be affected by disturbances of their physical environment (Schimel and Schaeffer, 2012). Garcia-Oliva et al. (2004) observed lower OC mineralization in crushed than in intact macro-aggregates and attributed this finding to a reduced microbial activity in crushed samples, what they explained by a disturbed soil environment with possibly anaerobic conditions. Balesdent et al. (2000) reviewed the effect of aggregate-crushing on the mineralization of soil OM and indicated a reduced microbial biomass in crushed aggregates as a possible reason for similar OC mineralization rates in intact and crushed aggregates. In our study, crushed macro-aggregates contained slightly but significantly less microbial biomass C than intact macro-aggregates (Fig. 6). This may have contributed to the missing effect of aggregate crushing on OC mineralization.

Besides stabilization of OC by physical occlusion within aggregates, formation of mineral-organic associations can be an important mechanism for OC stabilization (von Lützow et al., 2006). Bischoff et al. (2016) showed that a large OC fraction (>90% of total OC) is associated with mineral surfaces in soils of the Kulunda steppe, which is much more than



generally observed in steppe soils (Kalinina et al., 2011; Plante et al., 2010). In our study, about 38 ± 3% of the crushed macro-aggregate fraction were particles <63 μm, in which the proportion of particulate OC is usually very low (Christensen, 2001). Based on the similar OC mineralization rates of intact and crushed macro-aggregates, this suggests that a considerable OC proportion is stabilized by mineral surfaces. As a result, OC in crushed aggregates became not available to

5 microorganisms and thus did not enhance soil OC mineralization.

Summing up, our results suggest that the tillage-induced break-down of macro-aggregates and the subsequent release of OM is not the key factor driving OC losses due to LUC in the studied soils. In contrast, most OC in steppe soils of Siberia appears protected by occlusion within micro-aggregates and/or association with minerals. This, in part, contrasts with previous research of prairie soils from the North American Great Plains. Elliott (1986) and Cambardella and Elliott (1993,

1994) found that macro-aggregate occluded OC was rapidly lost after conversion of grassland to cropland due to the break-down of macro-aggregates and concluded that this fraction is protected from decomposition. Though, more recent research (e.g. Six et al., 2000) indicated that micro-aggregates which are formed within existing macro-aggregates are decisive for OC stabilization in agroecosystems, it is still widely accepted that the decomposition of previously occluded macro-aggregate OC is another key factor controlling the decline of OC after grassland to cropland conversion. Our results imply,

that this is not the case in the Siberian steppe soils. A possible explanation for the observed differences are smaller soil OC inputs by crop residues and rhizodeposits in the Siberian soils, resulting in smaller proportions of particulate OC (Castellano et al., 2015) and, thus, less possibilities for the formation of macro-aggregate occluded OC.

**Effect of land management and soil characteristics on the mineralization potential of soil organic carbon**

As shown in previous studies the conversion from grassland to arable land caused a decrease of labile soil OC (Plante et al.,

2011; Poeplau and Don, 2013), which corresponds to OC with fast turnover rates. In line with this, we found a larger fast OC pool under pasture than under arable land, while the proportion of the fast OC pool was unaffected by the intensity and duration of agricultural use (Fig. 4). This means, that the fast OC pool is highly vulnerable to LUC as the majority of this pool was rapidly lost within 1–5 yrs after grassland to cropland conversion. At the same time higher intensity or duration of agricultural land-use tended to shorten MRTs in the fast and slow OC pool of the soils in FS (Table 2), thus reducing the

potential to sequester soil OC. This is in line with Beare et al. (1994) and Grandy and Robertson (2007) who reported that the MRT of soil OC pools from laboratory incubations were shorter under high than under low land-use intensity. Beare et al. (1994) argued that the frequent soil disturbance in tilled soils impedes a strong association of OC with mineral surfaces, which in turn leads to a low protection of OC against microbial decomposition and thus fast turnover rates. Moreover, McLauchlan (2006) showed that the MRT of the fast OC pool was shorter in arable soils than in soils which were left as

fallow, which is supported by our results. We should consider that our observations were derived from long-term laboratory incubations and that we expect the difference of MRTs between arable and pasture soils to be even more pronounced under field conditions, as soil tillage generally accelerates the turnover of soil OC. Moreover, soil OC inputs by plant residues are





probably reduced in arable soils. This, together with faster soil OC turnover times would lead to a decrease of total soil OC as a result of agricultural land management.

We observed differences in the amount of mineralized soil OC between the two sites. Mineralization rates were smaller in the clayey soils of FS than in the soils of TS with larger sand content. Many studies showed smaller OC
mineralization rates in clayey soils as compared to sandy soils, as OC is stabilized by clay-sized minerals and thus protected against decomposition by microorganisms (Franzluebbers, 1999; Franzluebbers and Arshad, 1997; Harrison-Kirk et al., 2013). Moreover, Bischoff et al. (2016) showed that the proportion of labile particulate OC tended to increase with aridity in the soils under study. This means, that soils in TS would have larger amounts of bioavailable and easily decomposable particulate OC than soils in FS, which in turn leads to increased OC mineralization in the soils of TS. The differences
between both sites with respect to their soil OC mineralization rates could therefore be attributed to a different contribution of mineral-organic associations, with less mineral-bound OC in TS as compared to FS.

Interestingly, we found a smaller variation of the percentage of OC mineralized (i.e. OC mineralization rates) between samples from plots with long land-use duration (Fig. 2). This is possibly due to the fact that tillage homogenizes the soil within plough depth and consequently minimizes the heterogeneity of soil OC at the field scale. This idea is supported
by Schrumpf et al. (2011), who showed that soil OC contents are less variable under cropland as compared to grassland. We, therefore, conclude that continuous agricultural management obliterates differences of soil OC properties across a field.

**Conclusion**

This study set out to determine the quantity of macro-aggregate protected OC in Siberian steppe soils under different land-use and as function of land-use duration and intensity by crushing of dry-sieved macro-aggregates (250–2000 µm) to <250
20  µm and subsequent incubation of crushed and intact macro-aggregates at 20°C and 60% WHC during 401 days along two agricultural chronosequences of the Kulunda steppe. The effect of macro-aggregate crushing on OC mineralization was negligible along the two chronosequences. Macro-aggregate protected OC accounted for <1% of the total macro-aggregate OC content and for maximally 8 ± 4% of total mineralized OC. The majority of macro-aggregate protected OC was mineralized during the beginning of the incubation, showing that this represents a labile fraction with fast turnover rates. Our
results imply that the tillage-induced break-down of macro-aggregates has not reduced the OC contents in the studied soils. In contrast, our data suggest that mainly OC occluded within micro-aggregates and/or associated with mineral-surfaces is decisive for OC stabilization in these soils. Long-term incubations of bulk soil samples revealed that LUC from pasture to arable land but also the cultivation with forage crops caused a rapid decrease of a fast soil OC pool within 1–5 yrs of agricultural management. At the same time the MRT tended to become shorter in the fast and slow OC pool with increasing
land-use duration and intensity at one of the investigated sites. This suggests that the potential of the soils to sequester OC is reduced under agricultural management, as OC which enters the soil from above- or belowground is released to the atmosphere within few decades. The difference of turnover times between arable and pasture soils is probably even more





pronounced under field conditions, as soil tillage leads to a frequent disturbance of the soil environment which additionally accelerates soil OC mineralization. Thus, we conclude that the decrease of soil OC contents in the course of LUC is attributed to faster soil OC turnover under arable land as compared to pasture at a reduced plant residue input but not to the tillage-induced release of macro-aggregate occluded soil OC.

**Acknowledgements**

Financial support was provided by the German Federal Ministry of Education and Research (BMBF) in the framework of the KULUNDA project (01 LL 0905). O. Shibistova and G. Guggenberger appreciate funding from the Russian Ministry of Education and Science (No.14.B25.31.0031). We thank the entire KULUNDA team for great collaboration and good team spirit. We are thankful to all farmers of the Kulunda steppe for collaboration during sampling and Lukas Gerhard for
indispensable assistance in the field. Thanks for laboratory assistance to Silke Bokeloh, Elke Eichmann-Prusch, Roger-Michael Klatt, Pieter Wiese and Fabian Kalks, while Leopold Sauheitl is appreciated for guidance in the laboratory. Andrea Hartmann is acknowledged for helpful support on the scanning electron microscope. Thanks to Norman Gentsch for valuable scientific discussions on the manuscript, while we acknowledge Frank Schaarschmidt for statistical support. We thank an anonymous reviewer for valuable suggestions on the manuscript. The publication of this article was funded by the Open
Access fund of Leibniz Universität Hannover.



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





**Tables**

**Table 1: Land-use and soil properties in A horizons (pH, EC, sand, silt, clay) and in 0-10 cm (OC, TN, C : N) of investigated sites with mean annual temperature (MAT) and mean annual precipitation (MAP). Soil type classification according to IUSS Working Group WRB (2014). Abbreviation: n.d. = not determined.**

| Steppe | MAT | MAP | Soil type | Land use | pH | EC | Sand | Silt | Clay | OC | | | TN | | | C : N | | |
|---|---|---|---|---|---|---|---|---|---|---|---|---|---|---|---|---|---|---|
| | °C | mm | | | - | µS cm$^{-1}$ | mg g$^{-1}$ | mg g$^{-1}$ | mg g$^{-1}$ | mg g$^{-1}$ | | | mg g$^{-1}$ | | | - | | |
| Forest steppe | 1.1 | 368 | Protocalcic Chernozem | extensive pasture | 7.6 | 116.6 | 28 | 609 | 363 | 54.6 | ± | 5.4 | 4.5 | ± | 0.4 | 12.2 | ± | 0.2 |
| | | | | forage crop | n.d. | n.d. | n.d. | n.d. | n.d. | 49.3 | ± | 1.7 | 4.1 | ± | 0.2 | 12.1 | ± | 0.0 |
| | | | | arable 5 yr | 7.1 | 58.6 | 39 | 600 | 360 | 39.1 | ± | 1.4 | 3.3 | ± | 0.1 | 11.7 | ± | 0.1 |
| | | | | arable 30 yr | 7.0 | 60.4 | 34 | 598 | 369 | 40.4 | ± | 1.6 | 3.4 | ± | 0.1 | 11.8 | ± | 0.1 |
| Typical steppe | 2.0 | 339 | Protocalcic Kastanozem | fallow 30 yr (pasture) | 7.1 | 34.8 | 292 | 472 | 236 | 21.6 | ± | 2.3 | 2.0 | ± | 0.2 | 10.9 | ± | 0.1 |
| | | | | arable 1 yr | n.d. | n.d. | n.d. | n.d. | n.d. | 13.3 | ± | 0.3 | 1.3 | ± | 0.0 | 10.0 | ± | 0.1 |
| | | | | arable 3 yr | n.d. | n.d. | n.d. | n.d. | n.d. | 14.9 | ± | 1.6 | 1.5 | ± | 0.2 | 9.8 | ± | 0.2 |
| | | | | arable 10 yr | n.d. | n.d. | n.d. | n.d. | n.d. | 18.8 | ± | 1.5 | 1.8 | ± | 0.1 | 10.6 | ± | 0.2 |




**Table 2: Mean residence times of the fast and slow OC pool (years) as arithmetic mean ± SE, as derived from least-square fitting of incubation data, for two steppe types and as function of land-use and soil fraction. Significant differences (p<0.05) between fractions within land-use were not detected, which is indicated by same lowercase letters.**

| Steppe | Land-use | Fraction | Mean residence time (years) | | | |
|---|---|---|---|---|---|---|
| | | | Fast OC pool | | Slow OC pool | |
| Forest steppe | extensive pasture | bulk | 0.73 ± 0.07 a | | 62.8 ± 18.6 a | |
| | | intact | 0.74 ± 0.09 a | | 54.5 ± 1.6 a | |
| | | crushed | 0.69 ± 0.06 a | | 89.5 ± 19.2 a | |
| | forage crop | bulk | 0.46 ± 0.07 a | | 53.5 ± 7.6 a | |
| | | intact | 0.68 ± 0.11 a | | 65.2 ± 13.6 a | |
| | | crushed | 0.42 ± 0.06 a | | 48.3 ± 3.3 a | |
| | arable 5 yr | bulk | 0.51 ± 0.07 a | | 45.2 ± 8.0 a | |
| | | intact | 0.51 ± 0.06 a | | 48.1 ± 6.4 a | |
| | | crushed | 0.33 ± 0.02 a | | 42.4 ± 3.1 a | |
| | arable 30 yr | bulk | 0.38 ± 0.02 a | | 36.5 ± 1.6 a | |
| | | intact | 0.41 ± 0.03 a | | 41.5 ± 1.9 a | |
| | | crushed | 0.33 ± 0.01 a | | 36.9 ± 1.7 a | |
| Typical steppe | fallow 30 yr (pasture) | bulk | 0.79 ± 0.11 a | | 21.7 ± 7.1 a | |
| | | intact | 0.79 ± 0.10 a | | 25.8 ± 2.4 a | |
| | | crushed | 0.80 ± 0.10 a | | 25.9 ± 9.0 a | |
| | arable 1 yr | bulk | 0.33 ± 0.06 a | | 26.2 ± 4.9 a | |
| | | intact | 0.43 ± 0.05 a | | 30.9 ± 3.2 a | |
| | | crushed | 0.32 ± 0.05 a | | 30.3 ± 6.6 a | |
| | arable 3 yr | bulk | 0.68 ± 0.11 a | | 29.5 ± 4.0 a | |
| | | intact | 0.64 ± 0.14 a | | 20.6 ± 1.3 a | |
| | | crushed | 0.88 ± 0.16 a | | 22.7 ± 2.7 a | |
| | arable 10 yr | bulk | 0.55 ± 0.08 a | | 24.4 ± 1.4 a | |
| | | intact | 0.86 ± 0.18 a | | 33.2 ± 7.2 a | |
| | | crushed | 0.58 ± 0.10 a | | 25.9 ± 1.6 a | |



**Table 3: Proportion of macro-aggregate protected OC (% of mineralized OC) for two steppe types and the respective land-use. Since the proportion of macro-aggregate protected OC was calculated by subtracting the amount of OC mineralized in intact macro-aggregates from that in crushed macro-aggregates, negative values occur when the OC mineralization was smaller in crushed than in intact macro-aggregates.**

| Steppe | Land-use | Macro-aggregate protected OC (% of mineralized OC) | | |
|---|---|---|---|---|
| Forest steppe | extensive pasture | 1.4 | ± | 2.4 |
| | forage crop | -0.4 | ± | 8.4 |
| | arable 5 yr | 2.6 | ± | 4.3 |
| | arable 30 yr | 7.7 | ± | 4.2 |
| Typical steppe | fallow 30 yr (pasture) | 4.7 | ± | 0.8 |
| | arable 1 yr | 0.7 | ± | 1.8 |
| | arable 3 yr | -8.8 | ± | 5.7 |
| | arable 10 yr | 4.3 | ± | 3.1 |


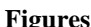

**Figures**

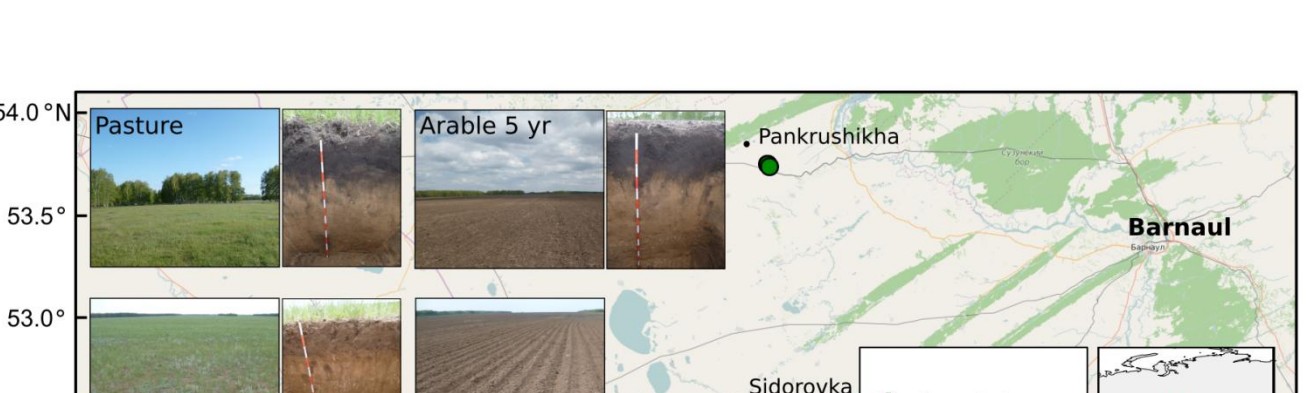

**Figure 1: Map of the study area and pictures from the two sites near Pankrushikha and Sidorovka. Map modified from**
5  **©OpenStreetMap contributors, for copyright see www.openstreetmap.org/copyright.**





**Figure 2: Percentage of soil OC remaining in the samples during 401 days of incubation for eight plots within two steppe types and for the three fractions bulk soil, intact macro-aggregates, and crushed macro-aggregates. For all plots a two-pool model (Eq. 2) was fitted, except for the extensively managed plots (extensive pasture and the fallow 30 yr (pasture)), where an asymptotic two-pool model was fitted (Eq. 3). Note the different scale for the fallow 30 yr (pasture).**





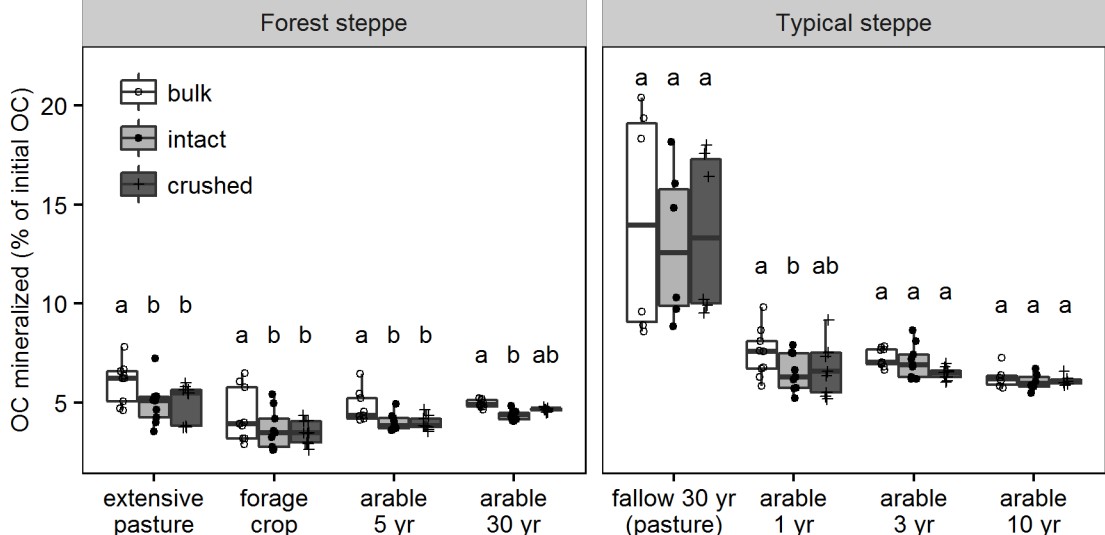

**Figure 3: Percentage of soil OC mineralized during 401 days of incubation for eight plots within two steppe types and for the three fractions bulk soil, intact macro-aggregates, and crushed macro-aggregates. Different lowercase letters indicate significant differences between fractions within plots at p<0.05.**





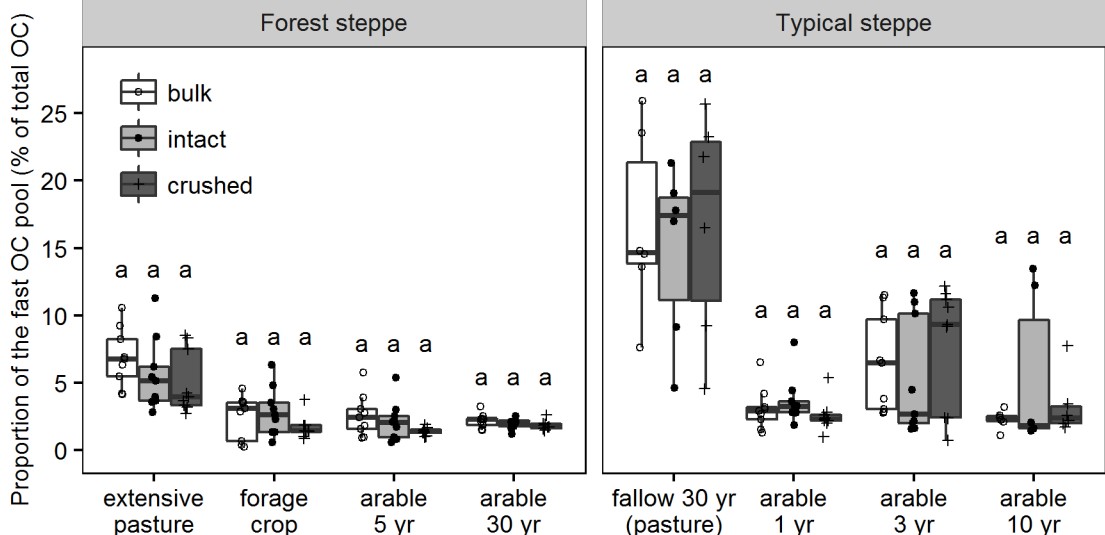

**Figure 4: Proportion of the fast OC pool (% of total OC) for eight plots within two steppe types and for the three fractions bulk soil, intact macro-aggregates, and crushed macro-aggregates as derived from two-pool model fits to incubation data (Eq. 2 and 3). Significant differences (p<0.05) between fractions within plots were not detected, which is indicated by same lowercase letters.**



**Figure 5: Mineralization rate of macro-aggregate protected OC (% of total mineralized OC d$^{-1}$) during 401 days of incubation for eight plots within two steppe types. The black solid line shows the mean mineralization rate per plot and the shaded grey area (confined by the black dashed lines) shows the corresponding standard error. The red dot-dashed line shows the fit of an exponential decay model (either 1-pool model, 2-pool model, or asymptotic 2-pool model according to the best fit). Since the mineralization rates were calculated by subtracting the OC mineralization rates of intact macro-aggregates from that of crushed macro-aggregates, negative values occur when the OC mineralization was smaller in crushed than in intact macro-aggregates.**




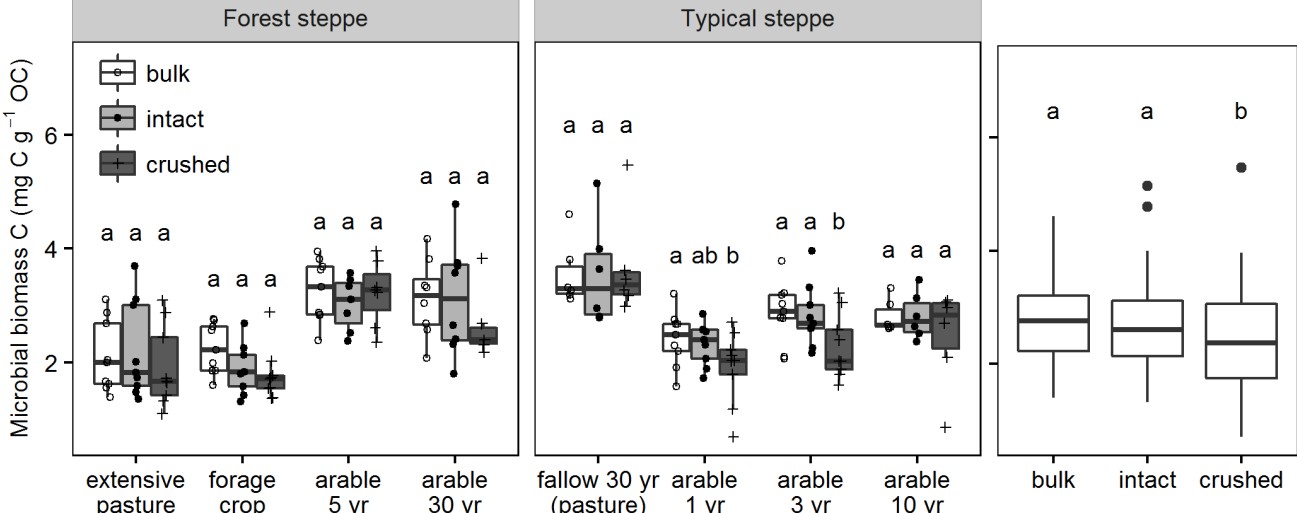

**Figure 6: Microbial biomass C (mg C g$^{-1}$ OC) for eight plots within two steppe types and the three fractions bulk soil, intact macro-aggregates, and crushed macro-aggregates. Different lowercase letters indicate significant differences between fractions within plots at p<0.05. The right panel shows differences between the three fractions averaged over all plots.**