# Peer review of "Limited protection of macro-aggregate occluded organic carbon in Siberian steppe soils"

_Biogeosciences, 2016_

## Referee Comment (RC1) · Anonymous Referee #1 · 18 Jan 2017

I have revised ms "Limited protection of macro-aggregate occluded organic matter in Siberian steppe soils". I have already provided these comments as a first revision, before the ms was published online, but found out that not all comments were properly reflected in the ms. Present work is a long-term laboratory incubation (400 d) of soils from two chronosequences (forest steppe and typical steppe). For each chronosequence soil plots with natural vegetation cover and agricultural management were chosen and samples were collected from the top horizon at the Kulunda steppe area (Russian Federation). From each soil, intact macroaggregates were separated by dry sieving, and the portion of them was crushed. The undisturbed soil samples, intact macroaggregates, and crushed macroaggregates were incubated with measurement of $CO_2$ emission rates, and microbial biomass in the end of the study. Results showed that crushing had only weak effect on the amount of mineralized C from macroaggregates,

and authors made a conclusion: 'macroaggregate occluded soil OM is not stabilized against decomposition in the Siberian steppe soils under study". Authors did a lot of work for maintaining such experiment, however, ms has several shortcomings, namely: i) the novelty of the study should be provided more clearly because making a study similar to other 10 studies, but with different objects, is not the novelty, ii) figures need to be redrawn because it is not clear what exactly they show (mean, median?), iii) result and discussion parts need to be rewritten with presentation only significant results. Please, see detailed comments below. For this stage, I advise major revision for the ms and suggest to strongly improve it.

General comments Please, make the sequential line numbering in all ms next time, and not in each page from line 1.

Instead of "aliquot" please use "subsample".

How did you fit the model to the experimental data for the $CO_2$ emission: for the each replication separately or for all replications together? If you have fitted the exponential model to all replication together, how did you then compare MRT of C pools between the plots (in this case you have only mean and st. error)?

How was the moisture controlled in during the experiment?

How can you be sure that aggregates were not formed again from crushed ones during incubation?

For the whole ms - be consistent with the terms - if you wrote arable, crop and pasture - use only them and not grasslands, croplands and so on. Use only one term - SOC or SOM. If you did not investigate N, use SOC only.

For MRT - it was longer/faster.

For the whole ms - write only about significant results, go through all ms and delete all sentences where "non-significant trends" and results are written. If you write only about significant results, please delete "(p<0.05)" everywhere in the text.

For the whole ms - do not repeat the numbers which are on the tables and on the figures, write about trends and make conclusions.

Discussion section - change the structure of the sentences - put the references in the end. Try to avoid sentences like: " The authors of the mentioned study suggested'...'.

Actually, the MRT is the most interesting part of the results, so you need to put more effort to discuss these results.

Specific comments P3 L 15-24 Actually this is already well known. Accordingly, Al-Kaisi et al. (2014) showed that macro-aggregates were faster disintegrated upon disturbance than micro-aggregates - This statement was already shown by Six et al., 2000.

P5 L5-10 - Put information about the MAP and MAT into the table 1 and delete from the text.

P5 L12-13, L24-25 The soil types are written in table 1, delete them from the text.

P5 L7 Add abbreviation for TS here.

P6 L6 Actually HCl fumigation is not the best method to remove carbonates from the soil, especially if you have high $CaCO_3$ content. Is it possible to check the initial $CaCO_3$ content and total C, to ensure that you have removed all $CaCO_3$?

P6 L18 It is not clear, did you incubate real field replications or analytical?

P6 L28 What was the reason to add $SiO_2$? You wrote to increase the volume, but maybe it was to prevent the formation of aggregates?

P9 L3-12 Completely delete. This information is presented in table 1, and you do not need to repeat it in the result section. Please write about trends.

P10 L16-21 Delete completely. This information is presented in table 2, you do not need to write these numbers again, write about trends and make the conclusions.

P13 L20 Fig 2 is a portion of remaining C and not mineralization rate.
Figures and tables Table 2 - It is not clear - is this whole soil, intact aggregates or crushed aggregates? You need to show all MRT if you have measured them.

Please look at the data for all of these categories, maybe these are differences?

Please add results of statistical tests for the MRT of fast OM pool between the plots.

Fig. 2 - Please present the mean and st errors for each sampling point for CO2, and not all experimental points.

Fig. 3 and 4 and 6 - I did not get, why you have tested the differences between the plots for all 3 fractions combined. Please provide differences for each fraction separately. Make a normal graph with mean and st. errors, what was the reason to make box plots and show the outliers?

Fig 4 Legend - This is not size, this is a portion of the fast mineralizable pool. Please, correct.

Fig 5 - Figure legend - this should be the "Mineralization rate of..." because your units are "% of total mineralized OC d-1", or there is a mistake in units. Make normal graphs - means (marked as dots), st. error to them and fitted exponential decay line. What does "Mean=1.4%" (and other similar information) mean? Mean mineralization rate? But it varies during a year.

Fig. 6 You can calculate % of MBC from the total SOC, and then you can compare plots between 2 chronosequences.
* * *

---

## Author Comment (AC1) · 18 Jan 2017

Dear Reviewer,

thank you for commenting on our manuscript and giving helpful advices to improve our manuscript. All of your comments from the first revision (before the ms was published online) were intensively discussed among the authors and a point-by-pont reply to your comments was submitted to the journal. In this reply, we indicate what we have changed in the ms according to your comments. We also show where the ms remained unchanged and argue why we would like to keep with the presentation style. However, it appears you have not been informed about this point-by-point reply. Therefore, we copy paste the complete reply to your first comments on the ms below (we also attach a reader-friendly PDF-version as a supplement to this comment –> see the hyperlink

below!). This reply covers all of your "new" comments from 18 January 2017. For example, we changed already the terms "aliquot" to "subsample" and referred to the MRT as being "shorter/longer". We also explained more clearly the novelty of the study and made respective changes in the ms. The reply was adressed to the Associate Editor (Y. Kuzyakov) but includes all of your comments.

Best regards,

Norbert Bischoff, on behalf of all co-authors

Reply to your first comments on the ms:

"Dear Prof. Dr. Yakov Kuzyakov, thank you very much for considering our manuscript for publication after minor revision. We intensively discussed the concerns raised by Reviewer#1 among co-authors and are pleased to present you a revised version of the manuscript. In this version, we better clarify the novelty of the study and point more precisely to the implications of our results (see for example P12 L6–17). We further improved the presentation of the data to facilitate the understanding for the reader (see for example Fig. 3–6 and Table 3). Moreover, we deleted unnecessary information from the main text (for example the results section "Soil organic carbon mineralization along the chronosequences" was completely rewritten) and focused on significant results and relevant trends to draw scientific conclusions (see for example P13 L24–27 or P14 L2–4). Thus, the revised manuscript is more concise. In the following, we respond step by step to each point raised by Reviewer#1. Best regards, Norbert Bischoff, on behalf of all co-authors

Reviewer#1 comments R#1: ms has several shortcomings, namely: i) the novelty of the study should be provided more clearly, because making a study similar to other 10 studies, but with different objects, is not the novelty, ii) figures need to be redrawn, because it is not clear what exactly they show (mean, median?), iii) result and discussion parts need to be rewritten with presentation only significant results. Please, see detailed comments below. For this stage, I advise major revision for the ms and suggest to strongly improve it. A: We thank Reviewer#1 for this critical comments. To the first (i) point: Research on the effect of land-use change from grassland to cropland on soil OM in steppe soils has, up to now, focused on the prairie soils (i.e. steppe soils) of the North American Great Plains. Very little is known about soil OC dynamics in the Siberian steppe soils, though this area belongs to the greatest grassland conversion areas of the world. Thus, studies about soil OC stabilization in the Siberian steppe biomes are quantitatively severely underrepresented. This is a critical imbalance and our study presented here is the first attempt to correct this (we summarized this problem already in the original manuscript P4 L6–9). Thereby, our study showed that the tillage-induced break-down of macro-aggregates and the subsequent release of OM is not the key factor driving OC losses due to land-use change in the Siberian soils. In contrast, most OC in steppe soils of Siberia appears protected by occlusion within micro-aggregates and/or association with minerals. This, in part, contrasts with previous research of prairie soils from the North American Great Plains. The here presented study is the first one documenting these differences between Siberian and American steppe soils and gives possible explanations to these findings. In our opinion, these are crucial novelties of our study. We more precisely point to that novelty in the revised manuscript and improved amongst others, the discussion section (P12 L6–17, P13 L24–27). Moreover, we indicated already in the original manuscript that a serious shortcoming of previous studies is the short incubation time. This complicates an accurate determination of the size of the macro-aggregate protected OC fraction and its turnover time. Our study is the first conducting a long-term incubation of intact and crushed macro-aggregates. By that, we could show that the fraction of macro-aggregate protected OC was negligibly small and that it was mineralized within 100–400 days. This approach, in fact, is a novelty and now pronounced more precisely in the revised manuscript (P4 L2–5). To the second (ii) point: It might be that some of our figures were not clear enough. To make clear what exactly they show, we defined the drawn boxplots in the statistics section of the revised manuscript (P8 L23–25). We went through the reviewer's suggestions regarding the figures (see below)

and incorporated many of these good advices. Part of the graphical design remained unchanged and we argue below in detail why we would like to keep with the presentation style. However, the helpful advices of the reviewer significantly improved the quality of the figures and the understanding for the reader. To the third (iii) point: We revised the discussion section and pointed towards the significant results. However, we keep some of the results which rather indicated trends than significant results. This concerns particularly the results relating to the MRT. First, Reviewer#1 argued that this is the most interesting part of the study, thus, we kept it in the manuscript. Second, differences of the MRT are mainly present between plots (and not between fractions). Owing to the experimental design, detecting significant differences between the plots is quite "difficult" as we have only three real field replicates per plot. Thus, all parameters which are compared between plots should be viewed as indicating trends rather than significant results. We satisfied this fact by removing statistical tests regarding differences between plots from the manuscript (see Fig. 3, 4, and 6 and Table 2) to not mislead the reader, and changed the statistical method description accordingly (P8 L2–8). This makes the manuscript more concise and the information presented in the figures and tables are easier to understand. Furthermore, in the revised manuscript we solely refer to the relevant trends.

General comments R#1: Please, make the sequential line numbering in all ms next time, and not in each page from line 1. A: This was done according to the template provided by the journal "Biogeosciences" for manuscript preparation. This can be seen also in articles of "Biogeosciences Discussions". Therefore, we have not changed this in the manuscript.

R#1: Instead of "aliquot" please use "subsample". A: Corrected.

R#1: How did you fit the model to the experimental data for the $CO_2$ emission: for the each replication separately or for all replications together? If you have fitted the exponential model to all replication together, how did you then compare MRT of C pools between the plots (in this case you have only mean and st. error)? A: We have

fitted the model to the experimental CO2 emission data for each replicate separately. This was already described in the original version of the manuscript (P7 L13–16). To better clarify this, we added "i.e. analytical replicate" in P7 L14.

R#1: How was the moisture controlled during the experiment? A: The moisture was controlled during the experiment by periodically weighing the incubation jars and adding the necessary amount of ultrapure water. This procedure is also described, for example, in Creamer et al. (2013). We added this information to the revised manuscript (P6 L29–30).

R#1: How can you be sure that aggregates were not formed again from crushed ones during incubation? A: As we mixed all samples (also those of crushed aggregates) with quartz powder, the probability of aggregate formation is very small. We added a respective note to the revised manuscript (P6 L27–28).

R#1: For whole ms - be consistent with the terms - if you wrote arable, crop and pasture - use only them and not grasslands, croplands and so on. Use only one term - SOC or SOM. If you did not investigate N, use SOC only. A: We changed the terms according to the comment. In the revised manuscript we use solely the terms "pasture" and "arable" land with respect to our studied plots. However, if we refer to results from other studies, we use the terms used in the cited study. For example, "grassland" refers in many studies to virgin grasslands and pastures. Moreover, we changed OM for OC in most cases throughout the manuscript. Also the title of the manuscript changed accordingly.

R#1: For MRT - it was longer/faster. A: We changed it accordingly to "the MRT is longer/shorter".

R#1: For whole ms - write only about significant results, go through all ms and delete all sentences where "non-significant trends" and results are written. If you write only about significant results, please delete "(p<0.05)" everywhere in the text. A: All of the "non-significant trends" referred to differences between the plots. As described above

(response to the third (iii) point), we revised our statistical approach and concluded that our experimental design did actually not allow for the detection of significant differences between plots, as we had only three field replicates in each plot. Our study was designed to detect statistical differences between soil fractions (bulk soil, intact, crushed macro-aggregates), as this was the main purpose of the study. Thus, we deleted all statistical tests relating to differences between plots. In the revised manuscript, we point towards the relevant trends of parameters (e.g. MRT, size of fast OC pool) across the plots of the chronosequences, and draw respective conclusions. Thus, we improved the understanding of the manuscript. When comparing parameters between soil fractions, we consider statistical significance and add "$p < 0.05$" to clarify at which significance level we detect differences. This is common when reporting statistical results and this was also done in many recent studies in the journal "Biogeosciences" (e.g. Gentsch et al., 2015; Hall et al., 2015; Schrumpf et al., 2013).

R#1: For whole ms - do not repeat the numbers which are on the tables and on the figures, write about trends and make conclusions. A: We deleted those parts of the manuscript where numbers on the tables and figures were solely repeated. For example, we completely revised the results sections "Soil organic carbon contents along the chronosequences" and "Soil organic carbon mineralization along the chronosequences".

R#1: Discussion section - change the structure of the sentences - put the references in the end. Try to avoid sentences like: "The authors of the mentioned study suggested'...'. Actually, the MRT is the most interesting part of the results, so you need to put more effort to discuss these results. A: In the revised manuscript, we list the references at the end of the sentences in most cases. Only in some cases we kept the references within a sentence, as they are placed behind the statements they represent. In the revised manuscript we changed sentences like "The authors of the mentioned study suggested'...' to better English, as we agree with the reviewer that these expressions were not optimal. The results about MRT are intensively discussed in our manuscript

in the discussion section "Effect of land management and soil characteristics on the mineralization potential of soil organic carbon" (in detail see P12 L23–32 and P13 L1–2). Conclusions about the MRT results and their implications are drawn in the conclusions section (P13 L29–32, P14 L1–4). Specific comments R#1: P3 L 15-24 Actually this is already well known. Accordingly, Al-Kaisi et al. (2014) showed that macro-aggregates were faster disintegrated upon disturbance than micro-aggregates - This statement was already shown by Six et al., 2000. A: We are well aware that this is already known and recapitulate the gained knowledge from previous studies in order to show that macro-aggregates are thought to play a crucial role for OC stabilization in the course of land-use change. This is important as it highlights the motivation of our study. We better clarified this in the revised manuscript and added three more references (P3 L23–25). The reference of Six et al. (2000) was a good advice and we changed the citation accordingly (see P3 L21–24 of the revised manuscript).

R#1: P5 L5-10 - Put information about the MAP and MAT into the table 1 and delete from the text. A: We now included MAP and MAT into Table1. However, we think it improves reading of the Material & Methods section if we remain the data about temperature and precipitation also in the text.

R#1: P5 L12-13, L24-25 The soil types are written in table 1, delete them from the text. A: As with the comment above, we believe that mentioning the soil types also in the text improves fluent reading and understanding of the experimental setup. Thus, we kept the soil types in the text.

R#1: P5 L7 Add abbreviation for TS here. A: The abbreviation was added as suggested.

R#1: P6 L6 Actually HCl fumigation is not the best method to remove carbonates from the soil, especially if you have high CaCO3 content. Is it possible to check the initial CaCO3 content and total C, to ensure that you have removed all CaCO3? A: The CaCO3 contents of A horizons of the investigated profiles were very small, typically

<0.1% of dry soil mass. Therefore, HCl fumigation was carried out to remove traces of CaCO3. Complete removal of CaCO3 was controlled via measurements of $\delta$13C of various subsamples. $\delta$13C ranged from -23.9 $\pm$ 0.2‰ to -26.3 $\pm$ 0.2‰ hence, indicating that carbonates were removed. If carbonates were present in the samples, $\delta$13C would show values closer to zero (Walthert et al., 2010). We added a sentence concerning this issue to the revised manuscript (P6 L6).

R#1: P6 L18 It is not clear, did you incubate real field replications or analytical? A: We agree that we have not sufficiently clarified the experimental design, which has now been corrected. We investigated 8 plots and took 3 field replicates per plot. All of these field replicates were divided into 3 fractions (bulk soil, intact, crushed). The 3 fractions, in turn, were divided into 3 analytical replicates. This gives a total of 216 samples (8 x 3 x 3 x 3). For clarification we added the information to the revised manuscript (P6 L17–19). Moreover, in the original version of the manuscript we already stated that we used three field replicates and remained this information in the revised manuscript (P5 L28–30).

R#1: P6 L28 What was the reason to add SiO2 ? You wrote to increase the volume, but maybe it was to prevent the formation of aggregates? A: Our primary intention of adding quartz to the samples was to increase the sample volume. Another aspect was, as the reviewer mentioned, that the formation of aggregates is prevented. We have added this to the revised manuscript (P6 L27–28).

R#1: P9 L3-12 Completely delete. This information is presented in table 1, and you do not need to repeat it in the result section. Please write about trends. A: We have not deleted this section completely, as it contains some important information (e.g., that increasing duration of agricultural land-use caused no further decrease of soil OC contents in arable soils of FS). However, we revised that section considerably and focused on the relevant trends presented in Table 1. Thus, the section became clearly shorter (P9 L3–8).

[Figure]

R#1: P10 L16-21 Delete completely. This information is presented in table 2, you do not need to write these numbers again, write about trends and make the conclusions. A: We deleted this part completely and have written it again, thereby focusing on the relevant trends. Moreover, in the revised manuscript we present the MRT of all three fractions in Table 2. Conclusions related to MRT data are made in the discussion and conclusion section (see P12 L23–32, P13 L1–2, P13 L29–32, P14 L1–4).

R#1: P13 L20 Fig 2 is a portion of remaining C and not mineralization rate. A: We agree with the reviewer that we wrongly referred to "OC mineralization rate" in Fig. 2, which has now been corrected. However, we added "i.e. OC mineralization rates" in brackets, as the different proportion of OC mineralized in the samples translates to different OC mineralization rates.

Figures and tables R#1: Table 2 - It is not clear - is this whole soil, intact aggregates or crushed aggregates? You need to show all MRT if you have measured them. Please look at the data for all of these categories, maybe these are differences?Please add results of statistical tests for the MRT of fast OM pool between the plots. A: In the original manuscript we solely showed the MRTs as function of land-use and not depending of the fractions as there were no significant differences between fractions. However, we agree with the reviewer that we should show the MRTs also of the fractions as their investigation is the main goal of the manuscript. Thus, we included the MRTs of all fractions in Table 2. We have not added the results of statistical tests between plots, as we renounced doing statistical tests between plots in the revised manuscript (see our response to the third (iii) point from reviewer#1).

R#1: Fig. 2 - Please present the mean and st errors for each sampling point for CO2 , and not all experimental points. A: We think that showing all experimental data points gives a better representation of the variability of the data set. Therefore, we remained with showing each replicate. By just plotting means and standard errors, the variability between the plots becomes not really apparent. This can be seen in the following figure (see Fig. 1 in the supplements to this comment), which is drawn in the suggested way.
R#1: Fig. 3 and 4 and 6 - I did not get, why you have tested the differences between the plots for all 3 fractions combined. Please provide differences for each fraction separately. Make a normal graph with mean and st. errors, what was the reason to make box plots and show the outliers? A: As stated in our previous comments, in the revised manuscript we do not present results from statistical tests between plots anymore. Thus, in Fig. 3, 4, and 6 we now solely indicate significant differences between fractions within plots. We decided to use boxplots as they give a better overview of the distribution and variability of the data than just plotting arithmetic means and standard errors. Another advantage of boxplots is that, normally, they show medians which are more robust against outliers than arithmetic means. Schädel et al. (2014), for example, used also boxplots in their analysis of incubation data. Therefore, we still use boxplots in the graphs of the revised manuscript. However, to clarify that medians are shown in the boxplots we added this information to the statistics section of the revised manuscript (P8 L23–25). R#1: Fig 4 Legend - This is not size, this is a portion of the fast mineralizable pool. Please correct. A: We corrected the figure accordingly.

R#1: Fig 5 - Figure legend - this should be the "Mineralization rate of..." because your units are "% of total mineralized OC d-1", or there is a mistake in units. Make normal graphs - means (marked as dots), st. error to them and fitted exponential decay line. What does "Mean=1.4%" (and other similar information) mean? Mean mineralization rate? But it varies during a year. A: We agree that it should correctly be termed "Mineralization rate of..." and changed it accordingly in the revised manuscript. Moreover, we deleted the text "Mean=1.4, SE=2.4%" etc. from the graph, as this information might be confusing here. This data ("Mean=1.4, SE=2.4%" etc.) refers to the share of the mineralized macro-aggregate protected OC in the totally mineralized OC during the entire incubation period. In the revised manuscript we have transferred this data to a separate table (Table 3). We also added some information to the caption of figure 5 to better understand the few negative values. However, we have not plotted means as dots, as by our statistical approach we could calculate the mineralization rate for each single day, thus, a continuous line would better represent the calculated data.

Also the standard error can be given for each single day and thus a shaded area better represents the calculated data than some error bars.

R#1: Fig. 6 You can calculate % of MBC from the total SOC, and then you can compare plots between 2 chronosequences. A:We implemented this advice in the revised manuscript.

References Creamer, C. A., Filley, T. R. and Boutton, T. W.: Long-term incubations of size and density separated soil fractions to inform soil organic carbon decay dynamics, Soil Biol. Biochem., 57, 496–503, doi:10.1016/j.soilbio.2012.09.007, 2013. Gentsch, N., Mikutta, R., Alves, R. J. E., Barta, J., Capek, P., Gittel, A., Hugelius, G., Kuhry, P., Lashchinskiy, N., Palmtag, J., Richter, A., Santruckova, H., Schnecker, J., Shibistova, O., Urich, T., Wild, B. and Guggenberger, G.: Storage and turnover of organic matter fractions in cryoturbated permafrost soils across the Siberian Arctic, Biogeosciences, 12, 4525–4542, doi:10.5194/bg-12-4525-2015, 2015. Hall, S. J., McNicol, G., Natake, T. and Silver, W. L.: Large fluxes and rapid turnover of mineral-associated carbon across topographic gradients in a humid tropical forest: Insights from paired 14C analysis, Biogeosciences, 12(8), 2471–2487, doi:10.5194/bg-12-2471-2015, 2015. Schädel, C., Schuur, E. A. G., Bracho, R., Elberling, B., Knoblauch, C., Lee, H., Luo, Y., Shaver, G. R. and Turetsky, M. R.: Circumpolar assessment of permafrost C quality and its vulnerability over time using long-term incubation data, Glob. Chang. Biol., 20(2), 641–652, doi:10.1111/gcb.12417, 2014. Schrumpf, M., Kaiser, K., Guggenberger, G., Persson, T., Kögel-Knabner, I. and Schulze, E.-D.: Storage and stability of organic carbon in soils as related to depth, occlusion within aggregates, and attachment to minerals, Biogeosciences, 10(3), 1675–1691, doi:10.5194/bg-10-1675-2013, 2013. Walthert, L., Graf, U., Kammer, A., Luster, J., Pezzotta, D., Zimmermann, S. and Hagedorn, F.: Determination of organic and inorganic carbon, $\delta$13 C, and nitrogen in soils containing carbonates after acid fumigation with HCl, J. Plant Nutr. Soil Sci., 173(2), 207–216, doi:10.1002/jpln.200900158, 2010.

[Figure]

Please also note the supplement to this comment:
http://www.biogeosciences-discuss.net/bg-2016-518/bg-2016-518-AC1-
supplement.pdf

———————————————————

[Figure]

Fig. 1.

---

## Referee Comment (RC2) · Anonymous Referee #2 · 2 Mar 2017

Page 2 Abstract

As a matter of fact, an abstract gives the paper (highly concentrated) consequently, all comments and recommendations given below apply also for the abstract.

Line 7: please add a reference to the statement "As the stabilization. . ...for maintaining soil fertility.

Line 9-11: in my understanding Lützow et al. do not point out that mineral-ass. and physical disconnection are the main ones, they rather want to strengthen that recalcitrance is not that important than thought

Line 11 13: delete this sentence, as your investigation is not done in the dry steppe,

rather substitute by own data (Bischoff 2016). (And to your knowledge, Kalinina et al. (2014) found comparable C rations in aggregate and clay fractions for dry steppe soils)

Line 16: is "primary particle" the right term? What about "detached" or "isolated"?

Line 22-23: I would add as explanation for the importance of aggregate C the very pronounced crumble structure (at least in Chernozems the best I have ever seen)

Line 1: what is meant by "complicate reliable assessment"? Please explain more detailed.

Line 5: please explain why Siberian steppe soils need special attention? Are results of the same soils but different regions not transferable? They must!

Line 15: delete "agricultural"

Line 18-20: first hypothesis is not consistently deduced from the literature! The authors state themselves that increases and decreases were found (page 3)

Line 20-21: also inconsistent: the authors refer to the opposite (page 3, line 29). The second part "land-use duration" and "intensity" (what is exactly meant by this term?) is not derived from knowledge from the literature (missing state of the art)

Line 24-25: is the approach of getting of getting pools from fittings decay models an appropriate one? Please explain to those who are not familiar with it, add references

Line1-2: As stated, the Kulanda steppe is semi-arid. How can FS be part of this steppe (forest steppe in semi-arid steppe?)?

Line 4: hopefully with comparable grain sizes within each chronosequence, please confirm

Line 5 (Tab. 1): I. missing data on grain size distribution, please add, so that any grain

size effect on analyzed process can be excluded. II. The term "soil type" is not used in WRB, please correct. III Replace the term "Typical Steppe" by "Semi-arid Steppe" and introduce abbreviation (throughout the text). IV here 30yr in line 12 ten years, what is correct?

Line 7: use one term throughout the text for "more arid typical steppe, you introduced before "semi-arid" (much more consistent) and be stay thereafter

Line 8: please clarify, how you identified sites

Line 9: I. Why did you resign to include a natural plot? All plots of second chronosequence have a management history, hence, no discussion on land use change can be done. Additionally, can you prove that the first plot (FS) has no cropping history at all, although cropped sites are that near and land use change seems to be distributed all over the investigation area , is crucial point (see discussion). II. The forage plot also has a time of cultivation, please add, otherwise infeasible. III What exactly makes the forage site to an intensively used one?

Line 13: abbreviation "TS" not introduced, should be exchanged by SAS (see above)

Line 19: "meanwhile"?, I guess since 1983

Line 16: 30 yr fallow means sampling year was 2013, correct? Please ad more information on sampling design

Line 22: "which are attributed to erosion" is a speculation, delete. E.g. position top hill vs. slope toe might also be feasible

Line 24: and if erosion is the case you have mixed material at plot 19 yr (autochthonous and from above) which makes plot 10 yr unfeasible. Strictly argued: Plot 10 yr has to be deleted, but what is left?

Line 25: How you prove that key profiles are representative?

Line 26: I. delete "genetic" II. arable 30yr plot not introduced before

Line28: despite not all sites were investigated by a key profile, all plots have to be analyzed in respect of grain size distribution (see above). II. Why EC was measured? Delete, if you do not refer to somewhere. III: what means "composite"? Mixed samples?

Line 30: Tab S1 not required, coordinates can be integrated in Tab 1

Line 2: which samples are "all samples"? Those from the profiles? And if so, why is always only one data set per soil is given, and not those per horizon? And from what horizon were the given data?

Line 3: why was the residual water measured? Nowhere appearing again

Line 18: statement on amount of samples is redundant

Line 20-25: it is not clear how the quantification was done. II. Fig "1 is not required, because 1) it does not help to understand the quantification and 2) does not appear again

Line 27: how was WHC determined?

Line 31: replace "sampling" by "filling"

Fig. 1: not introduced. II. site photos not meaningful (delete). III. profile too small. IV. Map not meaningful (medium scale is missing) + scale not stated + missing north arrow

Line7: what is meant by "increasing duration "? A reached equilibrium after 5 yr cropping?

Line 11: see comment on this issue above. In addition: at the top plot you might have include sub soil material as top soil was eroded. However, any erosion process includes addition from elsewhere or losses from top and addition from the sub soil, processes completely destroying investigation approach and hence have to be completely avoided (make sure). II. In addition, erosion statement was only given for 10 yr above, but not for 1 yr, as firstly described here.

Line 7 + 12: it might be interesting to point out the differences in C/N of both chronosequences?

Line 14 and following: it is not clear when the measurements were done, after incubation? It hampers a reviewing with regard to content

Line 16 (Fig S2): if the figure shall only show which is stated in the sentence beginning in line 14, the figure is not required (delete)

Line 17-19: I do not understand the relationship between the two sentences given here

Line 17 (Fig. 2): Figure does not show different scales, as written in the heading

Line 19-21 please interpret Fig 2 more correctly, what about TS?

Line 14-21: these few lines are supported by four figures/tables. I propose just to keep Fig. 3.

Line 23: advice to Fig. 3 needed but not to Fig 2 (delete)

Line 10: what is meant by "the sites", please indicate more precisely

Line 30 (Fig S3): not meaningful, delete

Line 32 (Fig S4): not meaningful, delete

Page 11 and 12 Discussion on limited protection macro-aggregate C

As already noted above I recommend a more carful discussion on this aspect. All plot of the second sequence have a cropping history. Thus, it might by possible that macro-aggregate occluded C was lost than, never built up again (Kalinina et al., 2011,

found during self-restoration of post-agrogenic Chernozems an increase in C, however in relations to other fractions increase of aggregate C was less existing) land thus, you find no differences. A cropping history can in all probability also not be excluded for the first plot of the first chronosequence. This means your chronosequences lack of proven uncropped former stage. So, once again this aspect has to be included into the discussion.

Page 12 and 13 Discussion on effect of management, soil and sites on mineralization

The discussion has to be done more tentatively, as differences in fast soil pools of grassland and cropland was not significant (see Fig. 4 and page 10, line 11). In this respect, statement beginning in line 28 is too offensive (just trends), the same is true for line 31 (or was this a literature statement, then add references). On the other hand the statement "our results support. . . . . . .." (page 13, line 5) is too general (what results are explicitly meant?). In addition be again be careful in discussion LUC from grass to crop (see comments for page 11, 12). At least, it is nice to see homogenous effects upon ploughing by your data, however, this is an old story.

Page 13-14 Conclusion

I again recommend a more careful writing many statement are not underlined by significant data, and again see comments for page 11, 12.

---

## Author Response (AR1)

Dear Prof. Dr. Yakov Kuzyakov,

thank you very much for considering our manuscript for publication after minor revision.

We discussed the comments given by Reviewer#2 among co-authors and are pleased to present you a revised version of the manuscript. In this version, we better present the state of current research on that topic and deduce the hypotheses more clearly. Moreover, the experimental design is explained more concisely. Many of the helpful advices of Reviewer#2 were incorporated into the manuscript. However, in some aspects we are still arguing against his/her major concerns. We were pleased about the many suggestions of Reviewer#2 and thank him/her for the critical discussion on our manuscript.

We would like to note, that Reviewer#2 commented on our originally submitted manuscript. After a first revision of the manuscript, according to comments of Reviewer#1, our manuscript was published in Biogeosciences Discussions. In that manuscript, many of the concerns of Reviewer#2 were already revised. That particularly concerns the discussion section and the abstract of the manuscript.

In the following, we respond step by step to each point raised by Reviewer#2. Those parts of the manuscript which were changed during the revision are indicated in parentheses (page, line) and refer to the marked-up version of the revised manuscript.

Best regards,

Norbert Bischoff, on behalf of all co-authors

**Reviewer#2 comments**

**R#2:** Page 2 Abstract. As a matter of fact, an abstract gives the paper (highly concentrated) consequently, all comments and recommendations given below apply also for the abstract.

**A:** Done as suggested.

**R#2:** Page 3. Line 7: please add a reference to the statement "As the stabilization. . ...for maintaining soil fertility".

**A:** Done as suggested.

**R#2:** Line 9-11: in my understanding Lützow et al. do not point out that mineral-ass. and physical disconnection are the main ones, they rather want to strengthen that recalcitrance is not that important than thought.

**A:** We changed the reference to Lehmann & Kleber (2015) who pointed towards the importance of mineral-associations and aggregates in their soil continuum model.

**R#2:** Line 11 13: delete this sentence, as your investigation is not done in the dry steppe, rather substitute by own data (Bischoff 2016). (And to your knowledge, Kalinina et al. (2014) found comparable C rations in aggregate and clay fractions for dry steppe soils)

**A:** We have not meant the "dry steppe" but wanted to pronounce that steppe ecosystems are "dry" and this lack of water might inhibit the formation of mineral-organic associations. Because of your comment we realized that this verbalization is confusing and therefore deleted "dry" from the sentence. We did not add our own data (Bischoff et al. 2016) in this sentence since this study has not explicitly detected the importance of aggregates for OC stabilization in steppe soils. In Bischoff et al. (2016) we rather found a relation between the soil OC decline and aggregate stability to which we refer on page 4 line 12.

**R#2:** Line 16: is "primary particle" the right term? What about "detached" or "isolated"?

**A:** We considered using "detached" instead of "primary", but came to the conclusion that "primary" is the better term. In the literature "primary particle" is used to describe particles which assemble aggregates. However, we deleted "free" as this term is not necessary here.

**R#2:** Line 22-23: I would add as explanation for the importance of aggregate C the very pronounced crumble structure (at least in Chernozems the best I have ever seen).

**A:** We did add the advice of Reviewer#2. We inserted this sentence on page 3 line 14-15 as we think it fits better there.

**R#2:** Page 4 Line 1: what is meant by "complicate reliable assessment"? Please explain more detailed.

**A:** The expression "complicate reliable assessment" was changed to: "which results in an unreliable assessment of the size of the macro-aggregate protected OC fraction and its turnover time". (page 4 line 5-6).

**R#2:** Line 5: please explain why Siberian steppe soils need special attention? Are results of the same soils but different regions not transferable? They must!

**A:** From a theoretical point of view results of the same soils but different regions should be transferable. But results of our previous study (Bischoff et al. 2016) showed that soils of the Kulunda steppe had different characteristics with respect to their OM quality, e.g. the partitioning between particulate OM and OM in mineral-organic associations, than soils of the North American prairies. Hence, despite soils of both regions are classified as Chernozems or

Kastanozems at level of soil groupings, differences in soil quality criteria might be expected. This suggests that soils in the Siberian steppes respond different to disturbances like land-use change than soils of the same soil groupings in the North American prairies. In fact, it is open whether results of soils of the same soil grouping are transferable between North America and Siberia. As to now, very little is known about the soil OC dynamics in the Siberian steppes, which is evident from the lack of studies in the literature.

**R#2:** Line 15: delete "agricultural"

**A:** We would not like to delete "agricultural" as this term specifies the fact that we were investigating chronosequences on agricultural land. The term "agricultural chronosequence" was e.g. also used by Insam and Domsch (1988) and Panettieri et al. (2014).

**R#2:** Line 18-20: first hypothesis is not consistently deduced from the literature! The authors state themselves that increases and decreases were found (page 3)

**A:** It is right that we state that increases and decreases were found. However, beforehand we deduce from the literature that OC is stabilized within aggregates. Moreover, we state that the disruption of aggregates was shown to be the reason for a decline of OC along agricultural chronosequences. These results/conclusions from the literature suggest that the disruption of macro-aggregates leads to an increased OC mineralization, as a previously occluded OC fraction becomes available to microbial decomposition. Thus, we made this to our first hypothesis. The question on why other studies have not found this increase is not part of our hypothesis, but is rather discussed in the Discussion part of our manuscript.

**R#2:** Line 20-21: also inconsistent: the authors refer to the opposite (page 3, line 29). The second part "land-use duration" and "intensity" (what is exactly meant by this term?) is not derived from knowledge from the literature (missing state of the art)

**A:** In the revised version of the manuscript we referred to "*bulk soil* OC mineralization rates (...) are higher in pasture than in arable soils" (page 4 line 22-23). This is not the opposite of what was written on page 3 line 29, as the statement in line 29 refers to the response of OC mineralization rates after aggregate crushing and not to OC mineralization rates of bulk soil. We agree with Reviewer#2 that we not precisely derived the second hypothesis ("land-use duration and intensity") from the literature. Therefore, we added a sentence clarifying the state of the current knowledge to the manuscript (page 3 line 27-29). The term "land use intensity" refers to the comparison of the pasture soil (low land use intensity) and the forage crop soil (relatively higher land use intensity) in the forest steppe. This is explained in the Material & Methods section (page 5 line 9-10). The term "land-use duration" refers to the "time since land-use change from pasture to arable land". To better clarify that, we changed the term and used it throughout the revised manuscript (page 4 line 20).

**R#2:** Line 24-25: is the approach of getting pools from fitting decay models an appropriate one? Please explain to those who are not familiar with it, add references

**A:** The use of double exponential decay models with two distinct carbon pools and associated mineralization rates constants is standard in describing the decomposition pattern of soil organic matter (e.g., Kalbitz et al., 2005). Therefore, in our opinion, it is not necessary to incorporate an explicit note on that topic into the manuscript. It is important that the time of incubation is sufficiently long (in most cases at minimum 1 year) that two C pools can be fitted accurately. Hence, we used an incubation time of >1 year.

**R#2:** Page 6 Line1-2: As stated, the Kulanda steppe is semi-arid. How can FS be part of this steppe (forest steppe in semi-arid steppe?)?

**A:** We thank Reviewer#2 for this attentive note and agree that a forest steppe cannot be part of a semi-arid steppe. Therefore, we deleted the term "semi-arid" from the manuscript.

**R#2:** Line 4: hopefully with comparable grain sizes within each chronosequence, please confirm.

**A:** Of course, the grain size of the soils was similar within one chronosequence. To avoid misunderstandings we added a respective sentence to the manuscript (page 5 line 4-5).

**R#2:** Line 5 (Tab. 1): I. missing data on grain size distribution, please add, so that any grain size effect on analyzed process can be excluded. II. The term "soil type" is not used in WRB, please correct. III Replace the term "Typical Steppe" by "Semi-arid Steppe" and introduce abbreviation (throughout the text). IV here 30yr in line 12 ten years, what is correct?

**A:** I. We added the necessary information on grain size distribution to Table 1. II. We deleted the term "type" and referred to "soil classification". III. We would not like to replace the term "Typical steppe" by "Semi-arid steppe" as (i) it is a characteristic term for the steppes of south-western Siberia and was already used in previous studies (e.g. Bischoff et al. (2016); Frühauf et al., 2004; Lebedeva (Verba) et al., 2008), and (ii) the term "semi-arid steppe" is broader and could also include the "dry steppe", which is located further south. IV. We thank the Reviewer for this attentive note. By mistake we wrote "ten years" in the main text, but "thirty years" is correct. This has been corrected.

**R#2:** Line 7: use one term throughout the text for "more arid typical steppe, you introduced before "semi-arid" (much more consistent) and be stay thereafter

**A:** In the previous comment we explained why we would not like to introduce the term "semi-arid" instead of "typical". Therefore, we keep the term "typical" and use it consistently throughout the manuscript. We agree that the expression "more arid typical steppe" can be confusing and changed the sentence in the revised manuscript (see page 5 line 8-9).

**R#2:** Line 8: please clarify, how you identified sites

**A:** We identified the two sites by interviewing farmers and land owners about land-use history and management. The plots within a site were checked for comparable pedological conditions by inspecting the soils with a hand-auger. We added this information to the manuscript to clarify how we identified sites (page 5 line 9-13).

**R#2:** Line 9: I. Why did you resign to include a natural plot? All plots of second chronosequence have a management history, hence, no discussion on land use change can be done. Additionally, can you prove that the first plot (FS) has no cropping history at all, although cropped sites are that near and land use change seems to be distributed all over the investigation area , is crucial point (see discussion). II. The forage plot also has a time of cultivation, please add, otherwise infeasible. III What exactly makes the forage site to an intensively used one?

**A:** I. We could not include a natural plot in our study, as we could not identify natural grasslands nearby our chronosequences. All grasslands in the region are normally used as extensive pastures. Nevertheless, it is possible to discuss effects of land-use change as the conversion of pasture to arable land, in fact, represents a change in land-use. Based on local farmers and land owners, we are very sure that the pasture plot in FS has no cropping history at all (at least not for the last ~100 years). This is underpinned by the fact, that there exists no relict/former plough horizon (Ap) which usually stays for decades once a soil was ploughed. II. The time of cultivation of the forage plot is 10 years, but we resigned to include this information into the manuscript, as this plot is not part of the chronosequence in FS, but rather used for the comparison of land-use intensity (pasture vs. forage crop). Therefore, the time of cultivation of the forage plot is not important for the interpretation of our results and we would not like to refer to it explicitly, to not create misunderstandings during reading of the manuscript. III. The forage crop plot is more intensively used than the pasture plot, as the cultivation of forage crops includes occasional soil management while soil management is absent on the pasture. The comparison of land-use intensity includes only the comparison between the pasture and forage crop in FS. This is already mentioned in the Material & Methods section of the manuscript (page 5 line 13-14).

**R#2:** Line 13: abbreviation "TS" not introduced, should be exchanged by SAS (see above)

**A:** In the version of the paper published in Biogeosciences Discussions the abbreviation "TS" was already introduced on page 5 line 7. As mentioned in a previous comment we would like

to keep the term "typical steppe (TS)" in the manuscript and not replace it by "semi-arid steppe (SAS)".

**R#2:** Line 19: "meanwhile"?, I guess since 1983

**A:** "Meanwhile" is the correct term. The plot was left as fallow since 1983 and is now used as pasture. Unfortunately, we do not know the exact year since it was used as pasture. Thus, it is "meanwhile" used as pasture. With respect to our experimental design it is not important since when it was used as pasture. The important fact is that it was not cropped and tilled since 1983.

**R#2:** Line 16: 30 yr fallow means sampling year was 2013, correct? Please add more information on sampling design

**A:** This is correct, sampling year was 2013. We added more information on sampling design to the manuscript (page 5 line 9-11).

**R#2:** Line 22: "which are attributed to erosion" is a speculation, delete. E.g. position top hill vs. slope toe might also be feasible

**A:** We would not like to delete the expression "which we attributed to erosion", to give the reader an idea why we did not measure a decline of soil OC along the chronosequence in TS, as is typical for chronosequence studies. Thereby, we point to the fact that it is very likely that another process superimposes the effect of soil management along the chronosequence in TS. It is important to note (and we included this sentence in the manuscript --> page 5 line 26-28), that the possible effect of macro-aggregate crushing on soil OC mineralization, if existent, will be also evident on slightly eroded plots. Therefore we decided to include the chronosequence in our study.

**R#2:** Line 24: and if erosion is the case you have mixed material at plot 19 yr (autochthonous and from above) which makes plot 10 yr unfeasible. Strictly argued: Plot 10 yr has to be deleted, but what is left?

**A:** We were discussing among co-authors to exclude the plot arable 10yr as it probably accumulated soil material from above (fallow 30 yr). Nevertheless, we decided to keep the plot in the study as it supports the general result of our study: "macro-aggregate protected OC is not stabilized against decomposition in the studied soils". Nevertheless, we agree that based on the chronosequence in TS there are no conclusions possible about the effect of land use duration. However, these effects are discussed based on the chronosequence in FS.

**R#2:** Line 25: How you prove that key profiles are representative?

**A:** We proved that by inspecting the soil with a hand-auger down to 1m depth.

**R#2:** Line 26: I. delete "genetic" II. arable 30yr plot not introduced before

**A:** We deleted "generic". II. We thank Reviewer#2 for this advice. As the reviewer correctly mentioned in a previous comment, we denoted the arable 30 yr erroneously as "10 yr" before, thus "arable 30 yr" was not introduced. We corrected that accordingly.

**R#2:** Line28: despite not all sites were investigated by a key profile, all plots have to be analyzed in respect of grain size distribution (see above). II. Why EC was measured? Delete, if you do not refer to somewhere. III: what means "composite"? Mixed samples?

**A:** Of course we checked all plots with respect of grain size distribution. On those plots where we did not establish a key profile we determined grain sizes by hand analysis and confirmed that all plots within a site had comparable grain size distribution. We added this information to the manuscript (page 6 line 1-4). II. Some of our colleagues argued that in steppe soils the electrical conductivity can vary considerably and hence affect microbial activity and in consequence OC mineralization. Therefore, we measured EC on those plots which were not located directly adjacent to each other. Our measurements show that in all plots EC was in a comparable range and confounding effects of EC on our results are unlikely. We agree with the Reviewer, that we did not refer to it elsewhere in the text. Thus, we decided to add a note in the results section of the main text (page 9 line 3-4). III. The term "composite" is not part of our manuscript, since it was deleted beforehand after a comment of Reviewer#1.

**R#2:** Line 30: Tab S1 not required, coordinates can be integrated in Tab 1

**A:** We agree with the Reviewer and included the coordinates in Table 1.

**R#2:** Line 2: which samples are "all samples"? Those from the profiles? And if so, why is always only one data set per soil is given, and not those per horizon? And from what horizon were the given data?

**A:** "All samples" refers to all incubated samples. We added the information to the manuscript to avoid misunderstandings (page 7 line 9).

**R#2:** Line 3: why was the residual water measured? Nowhere appearing again

**A:** This is a standard procedure in our laboratory and necessary to calculate the soil mass as basis for subsequent calculations of elemental contents. For example, if we measure the OC and TN contents (mg g$^{-1}$ soil) on air-dry samples we need to subtract the residual soil water content, otherwise we would underestimate the OC and TN contents. We added this information to the manuscript (page 6 line 9-10).

**R#2:** Line 18: statement on amount of samples is redundant

**A:** Since Reviewer#1 did not understand exactly the amount of samples, we clarified the quantity precisely in this sentence. We cannot find where this statement is redundant in the manuscript, as the sentence with the number "giving a total of 216 samples" is only given there.

**R#2:** Line 20-25: it is not clear how the quantification was done. II. Fig "1 is not required, because 1) it does not help to understand the quantification and 2) does not appear again

**A:** I. The quantification was done as following: the fraction of crushed macro-aggregates was sieved through a 63µm-sieve and the percentage remaining on the sieve and that passing the sieve was calculated by mass balance calculations. We added this information to the manuscript (page 6 line 28-29). II. In our opinion "Fig S1" is required as it highlights the condition of the crushed aggregates. Only by that figure we can conclude that the fraction <63µm consisted mainly of small micro-aggregates and only few primary particles (see page 6 line 31-32). Without "Fig S1" we could not rule out that the fraction <63µm is only composed of primary particles.

**R#2:** Line 27: how was WHC determined?

**A:** WHC was determined according to Schlichting et al. (1995). We added the information to the manuscript (page 7 line 2).

**R#2:** Line 31: replace "sampling" by "filling"

**A:** "Filling" is not right in this sentence as gas was sampled from the headspace of each jar and not filled.

**R#2:** Fig. 1: not introduced. II. site photos not meaningful (delete). III. profile too small. IV. Map not meaningful (medium scale is missing) + scale not stated + missing north arrow

**A:** I. We introduced "Fig. 1" on page 5 line 5. II. In our opinion the site photos are indeed meaningful and we would like to keep them in. In the site photos it is clearly visible that the

fallow 30 yr (pasture) in TS is degraded and has a sparse vegetation cover, to what we refer on page 5 line 22-23. This is in contrast to the "good" condition of the extensive pasture in FS, which is clearly visible in the site photos. III. We increased the size of the profile pictures. IV. In our opinion the map is meaningful as it quickly gives an overview to the reader where the study took place (without explicitly looking for the given geographical coordinates in Table 1 on a map). As we denote latitude and longitude in the figure a separate scale is not necessary. We added a north arrow to the figure.

**R#2:** Line7: what is meant by "increasing duration"? A reached equilibrium after 5 yr cropping?

**A:** We do not mean that an equilibrium was reached after 5 years of cropping. This could be a possible interpretation but we do not know about the soil OC content after >30 yrs cropping. Thus, we kept the expression "neutral" and solely stated that our data showed that the increasing duration of land use (pasture --> 5 yr arable --> 30 yr arable) has not led to a further decrease of soil OC contents.

**R#2:** Line 11: see comment on this issue above. In addition: at the top plot you might have include sub soil material as top soil was eroded. However, any erosion process includes addition from elsewhere or losses from top and addition from the sub soil, processes completely destroying investigation approach and hence have to be completely avoided (make sure). II. In addition, erosion statement was only given for 10 yr above, but not for 1 yr, as firstly described here.

**A:** I. No subsoil material was included at the top plot. As was denoted in the Material & Methods section (page 5 line 29) we took samples from 0-10 cm. In the profile picture of Figure 1 it is visible that the A horizon of the top plot (fallow 30 yr) was >20 cm, hence, subsoil material cannot be present when taking samples from 0-10 cm. In a previous comment we argued that we are aware that the possible erosion process along the chronosequence in TS will affect any interpretation regarding the effect of land use duration on bulk soil OC mineralization. Therefore, we interpret the effect of land use duration on bulk soil OC mineralization rates and sizes of the fast OC pool (hypothesis 2) solely based on results of the chronosequence in FS. However, the possible erosion process has no influence on the effect of macro-aggregate crushing on soil OC mineralization (hypothesis 1), as confirmed by the similar results of soil OC mineralization upon macro-aggregate crushing on the arable 10 yr plot. Thus, the general result of our study is not altered because of the possible erosion process. II. In this sentence we did not state that erosion took place on arable 1yr. We solely stated that "soil OC contents (...) did not follow the gradient over time since cultivation, as the site was affected by erosion". This means, that the arable 10 yr has not the smallest soil OC contents as we would have expected.

**R#2:** Line 7 + 12: it might be interesting to point out the differences in C/N of both chronosequences?

**A:** We pointed out the difference of C : N ratios between both chronosequences and included that topic in the discussion section (page 9 line 9, page 13 line 13-16).

**R#2:** Line 14 and following: it is not clear when the measurements were done, after incubation? It hampers a reviewing with regard to content

**A:** The measurement of OC and TN were done before the incubation experiment. We added this information to the Material & Methods section (page 6 line 13-14).

**R#2:** Line 16 (Fig S2): if the figure shall only show which is stated in the sentence beginning in line 14, the figure is not required (delete)

**A:** Fig. S2 indicates the respiration rates of the samples during the incubation. This figure is not necessary for the understanding of the manuscript. Therefore, we placed it in the supplements. In our opinion the figure should be kept in the supplements as it is a standard figure in incubation studies, giving an overview about the measured data (respiration/$CO_2$ emission of the studied samples), which otherwise cannot be given to the reader. We think that it is good style to show also primary data.

**R#2:** Line 17-19: I do not understand the relationship between the two sentences given here

**A:** We changed the formulation of the two sentences.

**R#2:** Line 17 (Fig. 2): Figure does not show different scales, as written in the heading

**A:** The figure does show different scales. Please note the different y-scale for the fallow 30 yr (pasture). We added "y-scale" to the manuscript to clarify that it is the y-scale and not x-scale.

**R#2:** Line 19-21 please interpret Fig 2 more correctly, what about TS?

**A:** Line 19-21 refer to Fig. 3 and not Fig. 2, as is mentioned in the text. We clarified this part of the text and indicated that the amount of soil OC mineralized was larger than that in the intact and crushed macro-aggregates in all plots along the two chronosequences (including TS), but significant differences were only detected in FS (page 9 line 21).

**R#2:** Line 14-21: these few lines are supported by four figures/tables. I propose just to keep Fig. 3.

**A:** We think that the four figures/tables are, in fact, not redundant and would like to keep them all in. Table S2 (in the revised manuscript Table S1) shows the absolute OC and TN content of the samples in mg g$^{-1}$ soil. Figure S2 indicates the original/measured incubation data and is in our opinion a standard graph in incubation studies which should be kept in the manuscript, though it is sufficient to place it in the supplements. Figure 2 highlights the fitted models. Based on these models we calculated the size of the fast and slow OC pool and the respective MRT´s. It is thus important that the reader can see how the models fitted the data and therefore it is necessary to keep Fig. 2 in the manuscript. Figure 3 summarizes statistics on data which is also present in Fig. 2, but could not have been integrated into Fig. 2 as that figure would become overloaded. Therefore, in agreement with Reviewer#2, Fig. 3 is also necessary to present the data to the reader in an appropriate way. Because of these reasons, we would like to keep all of the four figures/tables in the manuscript.

**R#2:** Line 23: advice to Fig. 3 needed but not to Fig 2 (delete)

**A:** See our comment above. However, in the revised manuscript we only refer to Fig. 3 and not Fig. 2, as Fig. 2 gives no information about the statistical significance of the results (page 9 line 24).

**R#2:** Line 10: what is meant by "the sites", please indicate more precisely

**A:** In the revised manuscript we do not refer to the sites anymore and indicated the results more precisely (page 10 line 11-12).

**R#2:** Line 30 (Fig S3): not meaningful, delete

**A:** In our opinion it is important to check for the relation between the percentage of OC mineralized and the quantity of soil microbial biomass, since soil microbes are the ones who respire and thus mineralize soil OC. We placed the figure to the supplements, as it does not indicate any of the main results. Nevertheless, it illustrates the reader the relation between the two parameters.

**R#2:** Line 32 (Fig S4): not meaningful, delete

**A:** In our opinion it is also important to check for the relation between the percentage of OC mineralized and the share of the microbial biomass C in the total soil OC. The share of the

microbial biomass C in the total soil OC was used by several authors as an indicator for soil OM quality (Allison et al., 2007; Hurisso et al., 2014), with larger values indicating a substrate with high OM quality. We would expect that a substrate with a high OM quality leads to larger OC mineralization rates, but this was not the case. As the figure did not show any of the main results we placed the figure in the supplements.

Page 11 and 12 Discussion on limited protection macro-aggregate C

**R#2:** As already noted above I recommend a more careful discussion on this aspect. All plots of the second sequence have a cropping history. Thus, it might by possible that macro-aggregate occluded C was lost than, never built up again (Kalinina et al., 2011, found during self-restoration of post-agrogenic Chernozems an increase in C, however in relations to other fractions increase of aggregate C was less existing) and thus, you find no differences. A cropping history can in all probability also not be excluded for the first plot of the first chronosequence. This means your chronosequences lack of proven uncropped former stage. So, once again this aspect has to be included into the discussion.

**A:** According to Kalinina et al. (2011) the increase of aggregate C (oPOM) after self-restoration of post-agronomic Chernozems was less pronounced than for other C fractions. But, particularly in 0-10 cm (the same sample depth as used in our study), Kalinina et al. (2011) observed an increase of oPOM to slightly >20% of total soil OC within 8 years, thus proving a build-up of aggregate C in the topsoil. Moreover, in our reply to a previous comment we clarified that we can be very sure that the extensive pasture in FS was never ploughed/cropped before. Hence, we can exclude that macro-aggregate occluded OC was already lost upon cultivation and never build up again on the extensive pasture in FS. The Reviewer is certainly right that we cannot prove this for the other plots as all of them were ploughed in former times. However, since the effect of macro-aggregate crushing on OC mineralization was not evident in the extensive pasture in FS, it is very unlikely that a missing build-up of previously lost macro-aggregate occluded OC is the reason that we found no differences in the other FS plots. In TS, the last ploughing of the fallow/pasture plot was 30 years ago and according to Kalinina et al. (2011) (Fig. 5 in their paper) aggregate-occluded OC increased by about 25–30% after 8 years of cultivation. Thus, we could expect a larger OC mineralization in crushed than in intact macro-aggregates at least in the fallow/pasture plot in TS, if a protection of macro-aggregate occluded OC was present. Since we did not find this increased OC mineralization, it is in our opinion correct to conclude, that macro-aggregate protected OC was not present in the studied soils. As explained in that comment, these conclusions remain even given the results from Kalinina et al. (2011). Apart from that, we changed many parts of the discussion on the effect of macro-aggregate crushing based on comments of Reviewer#1, and discussed the topic more carefully. Please refer to this formerly revised manuscript to see that changes.

Page 12 and 13 Discussion on effect of management, soil and sites on mineralization

**R#2:** The discussion has to be done more tentatively, as differences in fast soil pools of grassland and cropland was not significant (see Fig. 4 and page 10, line 11). In this respect, statement beginning in line 28 is too offensive (just trends), the same is true for line 31 (or was this a literature statement, then add references). On the other hand the statement "our results support. . .. . ..." (page 13, line 5) is too general (what results are explicitly meant?). In addition be again be careful in discussion LUC from grass to crop (see comments for page 11, 12). At least, it is nice to see homogenous effects upon ploughing by your data, however, this is an old story.

**A:** In the revised manuscript we deleted all statistical tests on parameters which were compared between plots, since our experimental design did actually not allow for powerful statistics on differences between plots (see our comments to Reviewer#1 which are published in the Biogeosciences Discussions forum). The experimental setup of our study was designed to test for significant differences between fractions (intact vs. crushed aggregates) to which we refer in the discussion section on the effect of macro-aggregate crushing. Since we removed statistical tests on differences of fast soil OC pools (see Fig. 4 and page 10 line 6-8) we had to discuss the results more tentatively. In our opinion, the statement in line 28 is not too offensive as it just expresses what we measured. In line 31 we denote that MRTs "tend" to become shorter along the chronosequence in FS. The statement "our results support..." was clarified in the revised manuscript (page 12 line 33). The effect of LUC from grassland to cropland is only discussed once in this section (page 12 line 25-26). There we state that the fast OC pool is highly vulnerable to LUC as it became diminished within 1-5 yrs after LUC. This is based on the results of the chronosequence in FS. As mentioned in a previous comment, the chronosequence in FS is valid and conclusions on the effect of LUC are therefore feasible.

Page 13-14 Conclusion

**R#2:** I again recommend a more careful writing many statement are not underlined by significant data, and again see comments for page 11, 12.

**A:** In the revised manuscript we changed many parts of the conclusion section in order to be more careful with the significance of the results. As we responded already on the comments for page 11 and 12, we would like to keep with our conclusion on the effect of macro-aggregate crushing, since in our opinion the results of Kalinina et al. (2011) do not question our results.

[revised manuscript text omitted]